# When Iterative RAG Beats Ideal Evidence: A Diagnostic Study in Scientific Multi-hop Question Answering

**Mahdi Astaraki**                                                      *astarakm@mcmaster.ca*
*Faculty of Engineering, McMaster University, Canada*
*BASF Canada Inc., Canada*

**Mohammad Arshi Saloot**                                   *mohammad.arshi-saloot@basf.com*
*BASF Canada Inc., Canada*

**Ali Shiraee Kasmaee**                                                  *shiraeea@mcmaster.ca*
*BASF Canada Inc., Canada*

**Hamidreza Mahyar**                                                    *mahyarh@mcmaster.ca*
*Faculty of Engineering, McMaster University, Canada*

**Soheila Samiee**                                                 *soheila.samiee@basf.com* [*]
*BASF Canada Inc., Canada*

**Reviewed on OpenReview:** *https://openreview.net/forum?id=pa5TnBdyDP*

## Abstract

Retrieval-Augmented Generation (RAG) is widely used to extend large language models (LLMs) beyond their parametric knowledge, yet it remains unclear when iterative retrieval-reasoning loops meaningfully outperform traditional static RAG, particularly in scientific domains where multi-hop reasoning, sparse domain knowledge, and heterogeneous evidence impose substantial complexity. This study provides the first controlled, mechanism level diagnostic evaluation of whether synchronized iterative retrieval and reasoning can surpass even an idealized static upper bound (Gold-Context) RAG in scientific domain. We benchmark eleven State-of-the-Art LLMs under three regimes: (i) No Context, measuring reliance on parametric memory; (ii) Gold Context, where all oracle evidence is supplied at once; and (iii) Iterative RAG, a training-free controller that alternates retrieval, hypothesis refinement, and evidence aware stopping. Using the chemistry focused ChemKGMultiHopQA dataset, we isolate questions requiring genuine retrieval and analyze model behavior through a comprehensive diagnostic suite covering retrieval coverage gaps, anchor carry drop, query quality, composition fidelity, and control calibration. Across models, iterative RAG consistently outperforms Gold Context, yielding gains up to 25.6 percentage points, particularly for non-reasoning fine-tuned models. Our analysis shows that synchronized retrieval and reasoning reduces late-hop failures, mitigates context overload, and enables dynamic correction of early hypothesis drift, benefits that static evidence cannot provide. However, we also identify limiting failure modes, including incomplete hop coverage, distractor latch trajectories, early stopping miscalibration, and high composition failure rates even with perfect retrieval. Overall, our results demonstrate that the *process* of staged retrieval is often more influential than the mere presence of ideal evidence in our evaluation experimental set up. We provide practical guidance for deploying and diagnosing RAG systems in specialized scientific settings and establish a foundation for developing more reliable, controllable iterative retrieval–reasoning frameworks. The code and evaluation results are available here.

---

[*]Corresponding author.

# 1   Introduction

Multi-hop question answering (QA) requires composing evidence across multiple steps and sources to arrive at a correct final answer. In scientific domains, this poses a particularly hard challenge: relevant knowledge is sparse, evidence must be chained across heterogeneous resources, and intermediate conclusions must be synthesized into final claims. As a result, multi-hop QA is a direct stress test of a model's ability to manage cognitive load, control deliberation, and integrate retrieval with reasoning (Adapala, 2025). Retrieval-Augmented Generation (RAG) has emerged as a central strategy to reduce reliance on parametric memory by grounding generation in external evidence (Lewis et al., 2020). Yet most evaluations treat retrieval as a static preprocessing step, followed by one-shot generation over a fixed context. Recent work argues that advanced RAG algorithms, such as *iterative* or *dynamic* RAG, can outperform static pipelines by progressively focusing the evidence set and correcting course mid-chain (Gao et al., 2025).

This strategy supports multi-hop QA in two complementary ways: (i) reasoning-augmented retrieval and (ii) retrieval-augmented reasoning. Prior work on reasoning-augmented retrieval typically assumes that the "ideal evidence", Gold Context supplied by dataset annotators, defines an upper bound, and thus evaluates how far improved retrieval can approach that bound without surpassing it (Nahid & Rafiei, 2025). Meanwhile, another group of studies (Xu et al., 2024; Li et al., 2025b; Wu et al., 2025) focus on enhancing retrieval-augmented reasoning, showing superior performance over direct reasoning without retrieval and over standard RAG pipelines. However, most existing comparisons use one step retrieval as the only baseline. This makes results highly sensitive to parsing, chunking, embedding, and re-ranking design choices, and obscures whether improvements stem from the algorithm or simply from retrieval configuration variance. Moreover, the Gold Context baseline (commonly included in reasoning-augmented retrieval studies) is often absent in evaluations of retrieval-augmented reasoning, making it difficult to form a complete picture. Although Gold Context is not guaranteed to be an *operational* upper bound, since it may be distracting for long chains with many internal hops, misaligned with a model's reasoning trajectory, or insufficient for compositional synthesis (Nahid & Rafiei, 2025; Chen et al., 2025), its inclusion remains essential for understanding the limits of static RAG.

Taken together, prior work tends to evaluate only one side of the retrieval–reasoning interaction—either retrieval-enhanced reasoning or reasoning-enhanced retrieval—and rarely examines how both critical baselines (No Context and Gold Context) jointly shape conclusions. Moreover, most studies evaluate only a small set of language models (typically fewer than five), which limits any systematic assessment of how model architecture influences observed performance. Existing survey papers primarily summarize reported results without offering deeper diagnostic insights, and direct, mechanism-level evaluation of retrieval-augmented reasoning in scientific multi-hop QA remains largely unexplored.

This paper offers a diagnostic re-examination of when and *why* synchronized retrieval and reasoning can beat ideal evidence in scientific multi-hop QA. Our goal is to jointly evaluate both aspects of potential enhancements within a single controlled framework, and to determine whether iterative retrieval can support reasoning strongly enough to surpass an idealized static evidence condition. To achieve this, we evaluate three regimes: (i) No Context (parametric memory only), (ii) Gold Context (Oracle evidence is supplied to the generator as a paragraph for each hop.), and (iii) Iterative RAG (a controlled retrieval - i.e., reasoning loop with explicit step allocation and stopping). Our study focuses on chemistry QA, a domain where general-purpose training provides limited coverage and where retrieval is genuinely required to bridge knowledge gaps. We begin with a No Context screen to remove questions answerable from internal memory and concentrate the analysis on retrieval-dependent cases.

We structure the investigation around four questions: (1) Accuracy: Under what conditions does iterative RAG outperform Gold Context, and how does this vary across model families and hop depths? (2) Utilization Dynamics: How do models use the retrieval loop to self-correct (e.g., anchor propagation), allocate steps across the chain, and calibrate stopping? (3) Failure Modes: What are the dominant sources of error in scientific multi-hop QA (e.g., coverage gaps in the final hop, composition failures despite sufficient evidence, distractor chains, and miscalibrated stopping)? (4) Efficiency and Compliance: What cost-accuracy archetypes emerge, and how strictly do models follow procedural constraints when parametric knowledge is tempting (*Procedural Compliance Rate*, PCR)? Rather than proposing a new dynamic-RAG algorithm or

producing a broad survey, we design a controlled evaluation that isolates the mechanisms by which synchronized retrieval and reasoning confer advantages – or fail. We compare more than ten language models spanning non-reasoning and reasoning-oriented architectures, track iterative utilization signals, and quantify sufficiency and coverage at each hop. To minimize configuration bias, we (i) standardize retrieval interfaces, (ii) decouple chunking and re-ranking from generation, and (iii) evaluate models under identical orchestration constraints.

Our results show substantial gains moving from parametric memory to Gold Context and, critically, further gains when synchronizing retrieval with reasoning. This study dissects the dual contributions of retrieval-enhanced reasoning and reasoning-enhanced retrieval, revealing that although the two interact, they benefit different model families and question structures in distinct ways. We illustrate that model architecture influences which mechanism contributes most, and that selecting the appropriate iterative RAG design for a given model can meaningfully reduce inference cost and latency while maintaining high precision. Notably, non-reasoning fine-tuned models gain disproportionately from structured retrieval-augmented reasoning, closing the performance gap with reasoning-specialized models and elevating smaller open-source models toward the performance range of much larger systems. In our evaluation, Iterative RAG consistently outperforms Gold Context baselines; for example, one frontier non-reasoning model achieves a +25.64 percentage point improvement in the iterative regime. We also find that certain newer models trade accuracy for efficiency, frequently bypassing the iterative loop and underperforming relative to earlier generations. Diagnostics reveal *Retrieval Coverage Gaps*, particularly at the final hop, *Composition Failures* where models fail to synthesize already-retrieved evidence, *Distractor Latches* anchoring chains to irrelevant facts, and *Miscalibrated Stopping.* Taken together, our evidence challenges the prevailing assumption that "more ideal evidence" is sufficient for multi-hop QA; instead, aligning retrieval with the reasoning trajectory is often the determining factor for success in specialized scientific settings.

To the best of our knowledge, this is the first diagnostic study demonstrating that iterative RAG can outperform ideal oracle evidence on a scientific multi-hop benchmark, and the first to explain *why* in terms of mechanism-level behavior. Beyond headline accuracy, our findings show that synchronized retrieval reduces late-hop failure cases and mitigates composition errors, while also highlighting that strict protocol adherence remains a challenge for frontier models. Our key *contributions* can be summarized as follows:

- A controlled, domain-specific diagnostic. We present a systematic comparison of eleven LLMs on multi-hop chemistry QA across No Context, Gold Context, and Iterative RAG, with common retrieval and orchestration controls.

- Evidence that iterative RAG can beat ideal evidence. We demonstrate that synchronized retrieval–reasoning can exceed strong Gold Context baselines, with the largest relative gains observed for non-reasoning models and deeper hop chains.

- Mechanism-level utilization analysis. We quantify how models leverage the retrieval loop to self-correct via anchor propagation, allocate steps across varying hop depths, and calibrate stopping.

- Failure mode taxonomy and impact. We characterize dominant failures, including final-hop coverage gaps, composition errors, distractor chains, and stopping miscalibration; and assess their impact on accuracy and generalization.

- Efficiency and adherence profiling. We identify cost-accuracy archetypes and measure *Procedural Compliance Rate* (PCR) to evaluate adherence to iterative protocols, especially when parametric knowledge is sufficient.

The rest of the paper is organized as follows. Section 2 reviews related work. Section 3 describes the experimental setup and diagnostic metrics. Section 4 reports comparative results across regimes, followed by deep analyses of failure modes and iterative utilization in Section 5. Section 6 discusses broader implications, and Section 7 concludes with recommendations for robust RAG in specialized domains.

## 2 Related Works

### 2.1 Datasets for Multi-hop Reasoning

Conventional RAG methods excel at factual question answering but face challenges when handling multi-step reasoning and complex decision-making tasks Gao et al. (2025). Research on multi-hop question answering has been driven by the release of several datasets designed to test reasoning beyond simple fact retrieval. HotpotQA Yang et al. (2018) is perhaps the most widely used example, with questions that force models to draw connections between different Wikipedia passages. Importantly, it also provides supporting sentences, which means systems can be judged on correctness but also on whether their reasoning steps are traceable. A related resource, 2WikiMultihopQA Ho et al. (2020), pushes this further by linking structured information from Wikidata with unstructured text, and by explicitly annotating inference paths, so that one can see how an answer ought to be derived.

Limitations of such Wikipedia-based datasets are that they are usually generic and also large language models (LLMs) may have already encountered portions of the data during pre-training. Consequently, the performance of a synergized LLM system on these public datasets may not accurately reflect its behavior on industrial domains, where the data is proprietary and unseen during pre-training. In more specialized settings, multi-hop reasoning shows up in other forms. For instance, financial question answering requires models to combine narrative text with tables and then carry out operations such as comparisons or basic arithmetic. This is the focus of FinQA (Chen et al., 2021) and TAT-QA (Yu et al., 2021), the latter being larger in scale and intentionally rich in numerical reasoning challenges. In the scientific domain, QASPER Dasigi et al. (2021) asks questions about research papers where answers may lie across different sections, meaning that retrieval alone is rarely sufficient. To resolve whether a scientific statement is supported, refuted, or left without evidence, SciFact Wadden et al. (2020) locate relevant material in multiple abstracts and weigh it together.

In chemistry, recent resources extend beyond generic multi-hop QA: ChemKGMultiHopQA introduces 1–4-hop, multi-document chains with KG supervision, while ChemLit-QA offers expert-validated literature questions tailored for RAG evaluation (Khodadad et al., 2025a; Wellawatte et al., 2025a). For benchmarking synergizing RAG and reasoning systems, a multi-hop QA dataset that features distinct intermediate answers and clearly defined linking entities provides the most effective evaluation framework. Although ChemLitQA-multi Wellawatte et al. (2025b) includes multi-hop questions, they are generally centered on a single shared entity connecting all hops. In contrast, the ChemKGMultiHopQA Khodadad et al. (2025b) dataset used in this study contains 1186 questions derived from ChemRxiv and enriched with information from PubChem and Wikipedia. It spans one to four hops and incorporates an automatically constructed knowledge graph with an expert-verified subset, enabling more comprehensive and diverse multi-hop chemical reasoning than both HotpotQA and ChemLitQA.

### 2.2 Iterative Retrieval-Augmented Generation

Synergized retrieval-augmented generation (RAG) and reasoning has received substantial recent attention (Gao et al., 2025; Li et al., 2025c; Trivedi et al., 2023; Tran et al., 2025; Shao et al., 2023), motivated by the need to bring large reasoning models closer to solving complex, real-world questions. Broadly, these methods fall into two complementary directions: reasoning-enhanced retrieval, where reasoning signals improve what to retrieve (e.g., Nahid & Rafiei, 2025; Chen et al., 2025; Cheng et al., 2024; Yan et al., 2025), and retrieval-enhanced reasoning, where retrieved evidence is repeatedly integrated to strengthen multi-step inference (e.g., Xu et al., 2024; Li et al., 2025b; Wu et al., 2025; Song et al., 2025a; Wang et al., 2025). In this landscape, iterative RAG instantiates synergy as explicit retrieve–reason–retrieve loops, where intermediate reasoning artifacts (plans, sub-questions, summaries, self-assessments, or verifiers) actively steer subsequent retrieval and synthesis.

A useful organizing lens is the procedural dynamism taxonomy of Gao et al. (2025), which contrasts pre-defined workflows (fixed pre-/post-retrieval reasoning hooks, or hybrid templates) with dynamic workflows in which retrieval and reasoning actions are conditionally triggered by the evolving problem state. Dynamic workflows tend to be superior in complex or open-world settings because they adapt to emergent task

complexity through state-contingent proactivity, reflection, and feedback loops, rather than following a rigid template. Concretely, iterative designs often rely on (i) planning and decomposition mechanisms that decide what to retrieve next and how to aggregate evidence, and (ii) explicit control signals that encode retrieval triggers, relevance, or verification decisions. For instance, PlanRAG follows an iterative plan-then-retrieve pattern: the model drafts a structured plan, issues targeted queries aligned with that plan, and re-plans as needed until sufficient evidence is gathered (Lee et al., 2024a;b). Complementarily, Self-RAG and SmartRAG use explicit prediction and control signals (e.g., special-token style decisions) to represent when to retrieve, whether evidence is relevant, and when to verify or revise (Asai et al., 2023; Gao et al., 2024).

Several recent methods highlight how iterative control and memory within the loop governs performance. ReSP targets multi-hop QA by recording the retrieval trajectory and using a dual-purpose summarizer to compress evidence with respect to both the global question and the current sub-question, mitigating context overload while supporting continued multi-step retrieval (Jiang et al., 2025). In domain settings with frequent follow-up needs, i-MedRAG operationalizes iteration as a sequence of LLM-generated follow-up questions: each is answered via a conventional RAG call, and the accumulated answers guide subsequent query generation, yielding an interpretable information-seeking chain (Xiong et al., 2024). Other work focuses on stopping and efficiency within the loop: Probing-RAG uses an auxiliary "prober" over intermediate representations to decide whether more retrieval is needed, reducing redundant steps while maintaining accuracy (Baek et al., 2025). More agentic frameworks emphasize evidence sufficiency: FAIR-RAG formalizes this by decomposing the question into required findings, identifying evidence gaps, and triggering targeted query refinement until the evidence set is deemed complete for faithful generation (Asl et al., 2025).

Across these approaches, controller design, meaning deciding when to continue retrieving versus finalize an answer, is repeatedly identified as a key driver of iterative gains, motivating hop-aware diagnostics and late-step gating in recent studies (Park et al., 2025a; Jiang et al., 2025; Chu et al., 2025). This is particularly salient in industrial and scientific use cases that require long-range, multi-step reasoning and multi-source evidence integration, where missing an intermediate retrieval step can break the logical chain (Gao et al., 2025).

## 3 Methodology

This section describes the evaluation framework used to test our main hypothesis. We first define the three evaluation regimes used to isolate parametric knowledge, oracle static evidence, and iterative retrieval–reasoning. We then describe the diagnostic suite used to attribute failures to retrieval, control, and synthesis. Finally, we introduce the benchmark, indexing procedure, iterative RAG controller, and answer-verification protocol.

### 3.1 Evaluation Framework and Metrics

We evaluate model performance across three distinct configurations to isolate the contributions of parametric memory, ideal evidence, and iterative reasoning. (i) **No Context:** The model answers the question relying solely on internal parametric knowledge, reflecting the internal knowledge gap of each model. (ii) **Gold Context:** An oracle text where the model receives the question and all ground truth paragraphs supporting every reasoning hop (the multi-hop question is generated from those paragraphs), with no further retrieval permitted. Synthesizing the upper bound of static retrieval then generation scenario, where the ground truth context is achieved in the retrieval attempt. (iii) **Iterative RAG:** The model utilizes our controller to actively retrieve evidence, refine hypotheses, and determine when to stop, subject to a step budget.

The Gold Context condition is designed as an idealized, noise-free static RAG baseline. In this setting, the model receives the minimum set of oracle supporting documents required to answer the question: the same paragraphs from which the multi-hop question was generated. These paragraphs therefore contain the answer path by construction, without retrieval noise or irrelevant distractors. In ChemKGMultiHopQA, each gold passage contains approximately 188 tokens on average, with a standard deviation of 56.26 tokens; therefore, even a four-hop question typically provides only about 750 tokens of oracle evidence. For this reason, Gold Context can be viewed as an optimal static retrieval condition: it assumes that the retriever

has already selected the necessary evidence with perfect precision, and the remaining challenge is evidence utilization and multi-hop composition.

The main question we ask is whether explicit test-time reasoning, independent of a model's prior reasoning-oriented fine-tuning, enables models to outperform an idealized static retrieval-generation setup on domain-specific multi-hop reasoning tasks. We further analyze how this effect varies across model families and hop depths.

**Difficulty Stratification.** To characterize model performance across complexity levels, we define questions as: ***Easy:*** Answered correctly by majority of models (2 or fewer models made mistake out of 11 tested models). ***Medium:*** Answered incorrectly by around half of the evaluated models (5 - 7 models). ***Hard:*** Answered incorrectly by majority of models (9 to 11).

### 3.1.1 Diagnostic Suite

To attribute failure modes to specific components (retrieval, reasoning, or control), we audit every iterative run using three families of diagnostics. Since the selected database provides the oracle hops, their path and gold context for each hop, in addition to the questions, answers and corpus; detailed evaluation of models' performance and their reasoning path is possible. All judgments are made strictly from the provided question, oracle hop path, and retrieval logs without using outside knowledge (See Prompts S8, S9, and S10.).

**I. Evidence Acquisition Diagnostics (Retrieval Quality)**

- **Retrieval Coverage Gap:** This metric is inspired by the sub-question coverage framework proposed by Xie et al. (2025), which evaluates whether retrieved chunks and the final answer collectively cover the facets (sub-questions) required by a complex query, and explicitly distinguishes failures due to missing retrieval evidence versus missing utilization of retrieved evidence. Building on this idea, we operationalize coverage at the level of oracle reasoning hops: for each oracle hop $k$, we check if *any* retrieved snippet at *any* step mentions the hop's key entity or relationship. If not, the hop is marked as *missed*. A missed hop indicates the retriever failed to fetch the necessary prerequisite for reasoning.

- **Query Quality Flags:** Multi-hop queries can draw on evidence from multiple steps, which helps reduce missing-information issues, but generating such queries directly with LLMs remains difficult (Shen et al., 2025). To analyze the semantic properties of the generated search intent, we classify each retrieval query using four mutually exclusive flags (For a detailed discussion of the impact, see Appendix S1.4):
    - *Vague:* The query lacks concrete targets (e.g., "learn more about HAT").
    - *Over-Broad:* The scope is too wide or mixes unrelated facets for the required hop.
    - *Fusion:* The query attempts to solve multiple oracle hops simultaneously in a single compound query.
    - *Off-Topic:* The query targets a subject not required by any oracle hop.

**II. Strategic Control Diagnostics (Planning & Adherence)**

- **Anchor Carry–Drop:** Prior multi-hop RAG work shows that intermediate retrieval and reasoning frequently drift away from intended subgoals ("unfaithful execution") and can become misaligned with retrieved evidence, motivating fine-grained, process-level evaluation beyond final-answer accuracy alone (Luo et al., 2026; Wei et al., 2025; Liu et al., 2025). We introduce Anchor Carry–Drop, a process-level drift indicator that detects whether a model preserves salient entities across hops—an essential prerequisite for stable multi-step reasoning in multi-hop RAG. From step $t > 1$, if the previous partial answer contains a salient anchor (e.g., a chemical formula), the subsequent query must carry at least one anchor. Failure to do so is flagged as a *carry–drop*, signaling a loss of working memory.

- **Confidence Miscalibration:** Following Yang et al. (2025), who define the core challenge in multi-round RAG as determining information sufficiency, namely deciding whether the currently retrieved information is adequate to answer the query or whether further retrieval rounds are required, we introduce Confidence Miscalibration to diagnose stopping logic. We diagnose stopping logic by flagging: (i) *Over-confident Finalize* (stopping before covering $\geq 80\%$ of hops despite having insufficient evidence), and (ii) *Under-confident Continue* (continuing retrieval despite having sufficient evidence).

- **Procedural Compliance Rate (PCR):** To complement Confidence Miscalibration, we introduce PCR to quantify adherence to the instructed verification protocol. It measures the proportion of known multi-hop questions (correctly answered in No Context) where the model actively verifies its answer by continuing the search beyond the mandatory minimum step ($Steps > 1$) rather than lazily finalizing at the first step.

- **Distractor Latch:** This is another process-level metric for measuring unfaithful execution and reasoning drift, capturing cases where the system fixates on near-miss terminology. Distractor Latch is a run-level failure where the system repeatedly locks onto a domain-specific similar but incorrect terminology. In the selected dataset, this manifests as a chemically similar but incorrect scaffold (e.g., retrieving "benzylic" instead of "phenoxyl").

### III. Information Synthesis Diagnostics

- **Composition Failure:** Recent multi-hop evaluations show that models can still answer incorrectly despite having sufficient retrieved evidence, and diagnostic analyses further report final-hop entity substitutions from entity confusion, motivating a metric that isolates synthesis failures from retrieval failures (Song et al., 2025b). Occurs when the correct entity/claim is present in the retrieved evidence, yet the final answer selects the wrong entity, provides a vague paraphrase, or merges competing entities. This isolates synthesis failures from retrieval failures.

- **Sufficiency Score ($\hat{s}$):** RAG evaluation frameworks emphasize claim-level faithfulness/attribution and explicitly distinguish retrieval misses from generator-side synthesis errors that occur even when supporting evidence is present in the retrieved context (Es et al., 2024). Sufficiency Score quantifies the fraction of sentences in the partial answers that are supported by at least one retrieved snippet. This measure reflects the extent to which generated claims are grounded in the provided evidence.

## 3.2 Experimental Setup

To perform our analysis, we implement a training-free, iterative retrieval-augmented framework designed to handle the heterogeneity and complexity of chemical reasoning. The system is composed of a specialized chemistry benchmark and a modular controller that alternates between retrieval and planning.

### 3.2.1 Dataset and Indexing

As explained earlier, to properly evaluate our hypothesis multiple criteria were required in identifying a suitable dataset for this experiment: multi-hop reasoning, domain specificity, and dependency on retrieval. These constraints made our search space relatively narrow. Consequently, benchmarks designed to evaluate reasoning capability independently of retrieval (e.g., OlympicArena (Huang et al., 2024), Humanity's Last Exam (Phan et al., 2025)) were excluded. Older benchmarks such as HotpotQA (Yang et al., 2018) were also not ideal, as they have been widely leaked into training corpora and contain only a limited subset of questions that genuinely require retrieval to address knowledge gaps. To minimize the impact of answer variability and dependency on writing style, we restricted our search to multi-hop QA datasets with short answers. Furthermore, to enable comprehensive failure mode analysis, we added the requirement that the dataset must provide all sub-questions, intermediate answers, and explicit connections between reasoning hops. Finally, we required the dataset to focus on a specific domain rather than general multi-hop QA. After applying these conditions, ChemKGMultihopQA Khodadad et al. (2025b) emerged as the benchmark

that best satisfied our requirements, and we selected it as the experimental testbed for this study. We also reviewed several recent multi-hop QA datasets from scientific, biomedical, legal, and open-domain settings, but none simultaneously satisfied the requirements needed for a controlled comparison across No Context, Gold Context, and Iterative RAG while also supporting mechanism-level diagnostics. We summarize this dataset-selection analysis in Appendix S1.6.

ChemKGMultiHopQA is a multi-hop chemistry dataset constructed from heterogeneous sources, including ChemRxiv, PubChem, and Wikipedia. We selected this dataset because it requires traversing distinct documents across one to four hops, ensuring that models must actively gather and compose evidence rather than perform single-step lookup. In addition, its short-answer format makes correctness verification more reliable and reduces dependence on style-sensitive evaluation metrics. The dataset is also recent, and prior results indicate relatively low No-Context accuracy, suggesting that many questions are difficult to answer without external evidence or explicit retrieval and reasoning. Finally, the availability of intermediate reasoning paths and hop-level annotations enables the detailed failure-mode diagnostics used in this study.

To prepare the corpus, we normalize text (Unicode canonicalization, whitespace cleanup) and segment documents into overlapping chunks of 220 words with a 50-word overlap. This window size is large enough to capture complete definitions or property mentions while avoiding the dilution of retrieval signals. We embed chunks using a chemistry-specific sentence encoder, `BASF-AI/ChEmbed` Kasmaee et al. (2025), which provides superior retrieval recall for scientific nomenclature compared to generic models.

### 3.2.2 Synergized Reasoning and Retrieval Implementation

Our system (Figure 1) functions as an orchestrator that coordinates three core responsibilities: *Retrieval* (finding evidence), *Planning* (deciding the next atomic step), and *Orchestration* (managing the loop). To enforce discipline, the process operates under a fixed budget of maximum 5 retrieval steps.

**Step Definition and Loop.** We define a single "step" as one distinct retrieval action. Therefore, a model that takes *1 step* performs exactly one retrieval query and immediately generates a final answer. A model that takes *5 steps* performs five sequential retrieval rounds before finalizing. The process begins with a mandatory first retrieval (Step 1) based on the user's initial question. Upon receiving the top-10 passages from Step 1, the planner enters the loop and must choose exactly one of two actions:

- **Retrieve (Step $K+1$):** If knowledge gaps remain, the planner formulates a new sub-query targeting the next logical hop (e.g., asking for a property of a compound identified in Step $K$).

- **Finalize:** If the planner decides that sufficient evidence has been gathered, it triggers a conservative composer that answers strictly from the provided passages with citations.

**Partial Answer State.** To maintain reasoning continuity, the system requires the generation of a *Partial Answer* before formulating a new query. We define the "Step $K$ Partial Answer" as the hypothesis generated *after* the model observes the query and passages from Step $K$. Although this generation technically occurs at the beginning of the prompt for Step $K+1$, it represents the state of knowledge derived from Step $K$. This summary acts as a communication channel (Yang et al., 2024), explicitly stating what has been confirmed to guide the subsequent retrieval.

**Context Management.** To prevent context dilution as steps increase, the planner receives a curated evidence view rather than the full history. At any given Step $K$, the model sees: 1. All passages from the *current* retrieval (Step $K$) in full (top-10). 2. A compact selection from previous steps (up to the 2 best passages from each Step $1 \ldots K-1$). This ensures that even at the step budget limit, the model processes a focused context of approximately 18 passages, preventing older information from overwhelming the latest evidence. Furthermore, partial answers and previous queries along with original question and the step number is passed to the model to make decision.

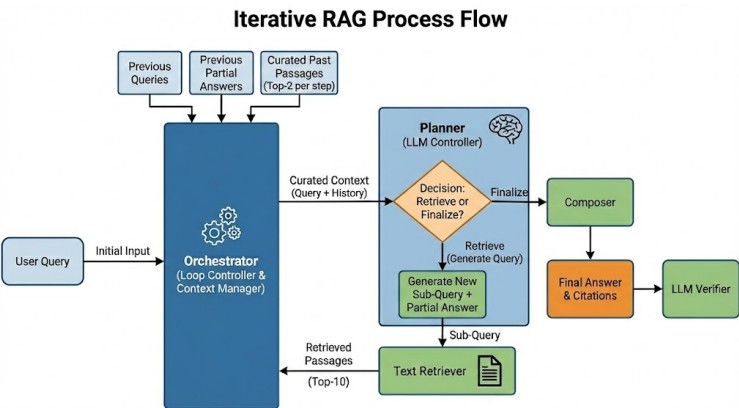

Figure 1: The Iterative RAG System: A training-free controller alternates between targeted retrieval and partial answer updates.

### 3.2.3 Correctness and Difficulty

Given the complexity of chemical nomenclature, exact string matching is insufficient. We verify answers using an LLM-as-a-judge (GPT-5-mini) protocol. The evaluation prompt (see S11) is designed to treat aliases, synonyms, molecular formulas, and IUPAC names as identical to the canonical answer. In the Iterative RAG setting, models may include brief explanations; such responses are considered correct if the final answer string is present as the specific final output. This verifier is the same verifier used in Khodadad et al. (2025b).

## 4 Results

Table 1 shows the results under three setups: No Context, Gold Context, and Iterative RAG. Figure 2 illustrates the Gaussian distributions of observed accuracy across these three setups aggregated over all evaluated models. Horizontal bars indicate pairwise $t$-test comparisons of accuracy, showing that the observed trends are consistent across models. Performance is weak in the No-Context condition ($37.16 \pm 5.54\%$). Providing Gold Context substantially improves accuracy ($69.14 \pm 7.22\%$). Iterative RAG, which couples retrieval with stepwise reasoning, yields further gains: ($80.89 \pm 5.08\%$); even the weakest Iterative RAG model, Llama 3.3 70B Instruct, performs nearly as well as the best Gold Context models, Gemini 2.5 Pro and Claude Sonnet 4.5, while the top Iterative RAG model, Claude Sonnet 4.5, surpasses the best Gold

Table 1: Per-mode accuracy and average output length (in tokens), reported by metric, under three conditions: parametric memory only (No Context), access to full oracle evidence (Gold Context), and Iterative RAG with synchronized retrieval and reasoning.

| Model | Accuracy (%) | | | Average Output Tokens | | |
|---|---|---|---|---|---|---|
| | No Ctx | Gold Ctx | Iter. RAG | No Ctx | Gold Ctx | Iter. RAG |
| OpenAI GPT-4o | 32.29 | 56.32 | 81.96 | **9** | **10** | 448.63 |
| OpenAI GPT-5 | **45.11** | 71.68 | 80.86 | 1565.48 | 713.41 | 5592.61 |
| Anthropic Claude 3.7 Sonnet (Reasoning) | 39.80 | 73.27 | 86.09 | 1777 | 715 | 4309.00 |
| Anthropic Claude 3.7 Sonnet (Standard) | 37.52 | 68.13 | 84.49 | 30 | 30 | 714.73 |
| DeepSeek R1 | 39.04 | 72.51 | 82.29 | 162 | 573 | 3671.38 |
| Mistral Large 2402 | 32.29 | 72.60 | 75.30 | 13 | 14 | 513.13 |
| Meta Llama 3.3 70B Instruct | 25.13 | 53.54 | 70.40 | 11 | 11 | **428.63** |
| Google Gemini 2.5 pro | 41.27 | **73.85** | 84.40 | 1733.71 | 1183.81 | 7214.59 |
| Anthropic Claude 4.5 Sonnet | 40.02 | **73.85** | **87.68** | 683.61 | 552.10 | 3524.80 |
| Z.ai GLM 4.6 | 35.49 | 72.51 | 78.67 | 3071.39 | 751.8 | 4303.75 |
| Grok 4 Fast | 40.85 | 72.26 | 77.66 | 2738.14 | 780.40 | 5292.99 |

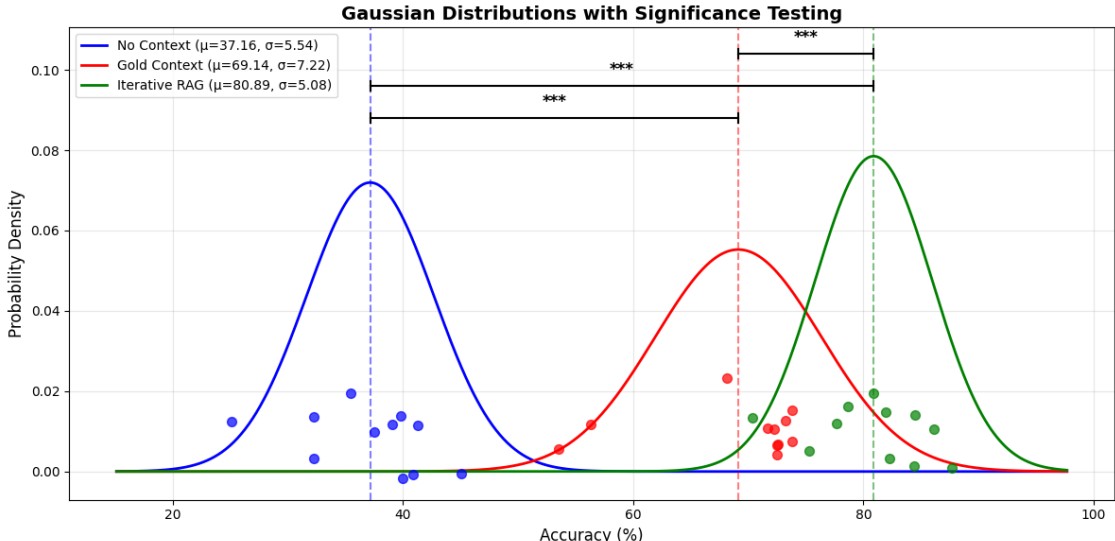

Figure 2: **Models' Accuracy: Distribution of Model Accuracy Across No Context, Gold Context, and Iterative RAG.** No Context, Gold Context, and Iterative RAG are shown in blue, red, and green, respectively. Horizontal bars show the results of pairwise t-tests (Significance: *** $p < 0.001$, ** $p < 0.01$).

Context result by 13.83 percentage points. All differences are reported in percentage points (pp) in this study.

Moving from the parametric memory baseline (No Context) to ideal retrieval (Gold Context) yields substantial accuracy gains across all architectures and training setups. This reflects the importance of retrieval in models performance in this scientific multi-hop reasoning task. Synchronizing retrieval and reasoning (iterative RAG), yielded even more dramatic performance shift, consistently outperforming the Gold Context static retrieval setup. For instance, GPT-4o (OpenAI, 2024b) jumps from 32.29% (No Context) to 81.96% (Iterative RAG), a massive 49.67 pp gain, significantly exceeding the 24.03 pp gain achieved by providing Gold Context. Similarly, Claude Sonnet 4.5 sees its performance more than double, rising from 40.02% to 87.68%. This indicates that for complex multi-hop tasks, the *process* of active information retrieval and stepwise state tracking is often more valuable than being provided with the correct supporting passages achievable in an ideal static RAG.

When isolating the specific gains from Gold Context to Iterative RAG, we observe distinct behaviors. On average, non-reasoning models gain 15.39%, whereas reasoning models gain 9.67%. However, the non-reasoning group exhibits high variance: GPT-4o achieves the largest relative gain of 25.64%, while Mistral Large (AI, 2024c) shows the smallest increase (2.70%). This suggests that iterative retrieval acts as a critical scaffold for models with weaker static context utilization (like GPT-4o), whereas strong non-reasoning models like Mistral Large maximize the Gold Context effectively, leaving less marginal utility for the iterative process. Nevertheless, reasoning-optimized models still derive consistent value; for example, Claude Sonnet 4.5 improves by 13.83 pp, confirming that structured iteration helps narrow the search space even for the most capable reasoning engines.

With respect to token usage, reasoning models typically produce more output tokens in the baseline condition than in ideal retrieval setup, as they externalize reasoning to compensate for missing evidence. In some cases, particularly with (ZhipuAI, 2025), responses exceed reasoning token limits (8196), yielding no final answer. Iterative RAG requires substantially more tokens: up to five retrieval steps are available, and each step generates both a query and a partial answer that serves as cross step state; consequently, output tokens and cost increase. Overall, No Context uses roughly twice the output tokens of Gold Context, and Iterative RAG uses about three times those of No Context on average.

GLM, as noted earlier, produces the most output tokens in the No-Context setting. In Gold Context and Iterative RAG, Gemini 2.5 Pro exhibits the highest average output tokens. This reflects a long form response style that persists even when high-quality context is available.

Among reasoning optimized models, Gemini 2.5 Pro yields the highest overall token counts, while GLM dominates in No Context. In contrast, DeepSeek R1 has the lowest token usage in No Context, and Claude Sonnet 4.5 consumes the lowest number of tokens in both Gold Context and Iterative RAG.

For non-reasoning models, the No-Context and Gold-Context settings typically require only the final answer string; by design, Iterative RAG additionally permits partial answers and cross step state, which increases token usage. Within this group, Llama uses the fewest tokens, whereas Claude 3.7 Sonnet (Standard) uses the most. This increase is expected, as iterative setups trade tokens for improved grounding and accuracy.

## 4.1 Gold+CoT Ablation: Is the Iterative Gain Merely More Computation?

One possible explanation for the advantage of Iterative RAG over Gold Context is that the iterative setting allows the model to produce substantially more output tokens. In this view, the observed improvement may not come from staged retrieval itself, but simply from giving the model more opportunity to reason before producing the final answer. To test this possibility, we introduce an additional compute-augmented static baseline, denoted **Gold+CoT**. In this setting, the model receives the same oracle evidence as in the Gold Context condition, but is explicitly prompted to reason step by step before producing the final answer. No additional retrieval is allowed, and all oracle paragraphs are still provided simultaneously. Thus, Gold+CoT preserves the static-evidence structure of Gold Context while giving the model a larger reasoning budget.

Table 2 compares Gold Context, Gold+CoT, and Iterative RAG for three representative models. Gold+CoT improves over the standard Gold Context baseline for all evaluated models, confirming that explicit test-time reasoning is useful in scientific multi-hop QA. However, Gold+CoT still remains below Iterative RAG in all cases. For GPT-4o, Gold+CoT increases accuracy from 56.32% to 78.08%, but Iterative RAG further improves performance to 81.96%. A similar pattern holds for Claude 3.7 Sonnet (Standard), where Gold+CoT reaches 81.45%, while Iterative RAG reaches 84.49%. Importantly, this improvement cannot be explained by token count alone: Claude 3.7 Sonnet (Standard) uses more output tokens in Gold+CoT than in Iterative RAG (909.96 vs. 714.73), yet still performs worse. This indicates that simply allowing longer reasoning over static oracle evidence is not sufficient to match the benefit of synchronized retrieval and reasoning.

Table 2: Gold+CoT ablation comparing static oracle evidence with additional reasoning against Iterative RAG. Accuracy is reported in percentage points, and token counts correspond to average output tokens. Gold+CoT improves over Gold Context, but Iterative RAG remains more accurate across all evaluated models, suggesting that the gain is not merely due to greater output length or additional computation.

| Model | Gold Context | | Gold+CoT | | Iterative RAG | |
|---|---|---|---|---|---|---|
| | Acc. | Tokens | Acc. | Tokens | Acc. | Tokens |
| GPT-4o | 56.32 | 10.00 | 78.08 | 262.10 | **81.96** | 448.63 |
| Claude 3.7 Sonnet (Standard) | 68.13 | 30.00 | 81.45 | 909.96 | **84.49** | 714.73 |
| Claude 3.7 Sonnet (Thinking) | 73.27 | 715.00 | 83.22 | 2133.52 | **86.09** | 4309.00 |

These results suggest that the advantage of Iterative RAG is not merely a consequence of using more computation. Additional reasoning over the Gold Context closes a substantial portion of the gap, showing that computation matters; however, Iterative RAG still achieves the highest accuracy across all three representative models. The key distinction is how that computation is organized. In the iterative setting, extra tokens are spent on generating targeted search queries, maintaining partial answer states, and conditioning each reasoning step on newly retrieved evidence. This staged structure changes the reasoning process itself: instead of forcing the model to synthesize all oracle evidence in a single static pass, Iterative RAG repeatedly narrows the evidence space, updates the working hypothesis, and aligns each retrieval action with the current reasoning state. Therefore, the key benefit is not longer generation alone, but the synchronization

of evidence acquisition and reasoning across multiple steps. The exact Gold+CoT prompting template is provided in Appendix Figure S12.

## 4.2 Decomposing Performance: The "Synchronized" Advantage

While aggregate metrics indicate strong performance, Figure 3 decomposes the problem space into four mutually exclusive categories to reveal *how* each model achieves its final accuracy. The rows sum to 100%, partitioning the dataset into: (i) **Parametric Memory Wins** (solved by internal knowledge alone), (ii) **Optimum Retrieval Wins** (unlocked only when ideal evidence is provided statically), (iii) **Synchronized Retrieval and Reasoning Wins** (solved *only* via Iterative RAG, failing under both No Context and Gold Context), and (iv) **Not Solved** (failed in all settings).

This partition reveals a critical finding: Iterative RAG does not simply subsume Gold Context; it solves a distinct class of problems. The "Synchronized Retrieval" column represents questions where static evidence is insufficient, and the model explicitly requires stepwise navigation to derive the answer.

- **High-Value Scaffolding for Non-Reasoning Models:** Non-reasoning models, such as LLAMA 3.3 70B and GPT-4O, exhibit the largest "Iterative-Exclusive" shares (25.6% and 27.8% respectively). For these models, the iterative process acts as a necessary scaffold, allowing them to solve over a quarter of the dataset that they could not handle even with perfect static evidence.

- **Hidden Volatility in Strong Models:** The figure exposes trade-offs hidden by the net accuracy scores. For instance, MISTRAL LARGE 2402 shows 13.8% in the "Iterative-Exclusive" column, meaning it successfully reasoned through many hard problems that Gold Context failed to solve. However, its net gain in Table 1 is only about 2.7%. This discrepancy implies a high regression rate: while iteration enables the model to solve 13.8% of new hard cases, it simultaneously loses the ability to solve approximately 11% of cases that could have been addressed in a single static run under ideal retrieval conditions.

Thus, Figure 3 clarifies that the benefit of Iterative RAG is orthogonal to Gold Context: it unlocks complex reasoning chains (Column 3) but introduces a stability risk that varies by model. The share of *not solved* questions also varies appreciably across models. CLAUDE SONNET 4.5 and GEMINI 2.5 PRO leave relatively few questions unresolved (about 7–8%), whereas LLAMA 3.3 70B INSTRUCT, GROK 4 FAST, and GPT-4O retain larger unsolved portions ($\approx$11–18%).

## 4.3 Stability Analysis: Recoveries vs. Regressions

To investigate the volatility identified in the solvability partition, we analyze the specific trade-offs between static and dynamic setups. As shown previously in Table 1, all models benefit from Iterative RAG relative to both No Context and Gold Context. Nevertheless, iteration does not dominate everywhere: there remains a subset of questions answered correctly under Gold but not under Iterative RAG. Figure 4 quantifies this trade-off by contrasting *recoveries* (Gold incorrect $\rightarrow$ Iterative correct) with *regressions* (Gold correct $\rightarrow$ Iterative incorrect).

MISTRAL LARGE 2402 exhibits many recoveries but also many regressions, yielding the lowest net gain (approximately +33). In contrast, the other non-reasoning models GPT–4O and LLAMA 3.3 70B INSTRUCT achieve the largest net gains, followed by CLAUDE 3.7 SONNET (standard). Among all models, CLAUDE SONNET 4.5 has the fewest regressions. The dominant pattern is that iteration *recovers* many questions that static Gold Context fails to resolve, confirming that interleaving retrieval with reasoning adds capability rather than merely adding more text.

## 4.4 The Cost of Retrieval: Parametric Memory Suppression

A specific and counter-intuitive subset of regressions occurs when the introduction of retrieval actively degrades performance on questions the model already "knows." We term this phenomenon *Parametric Memory*

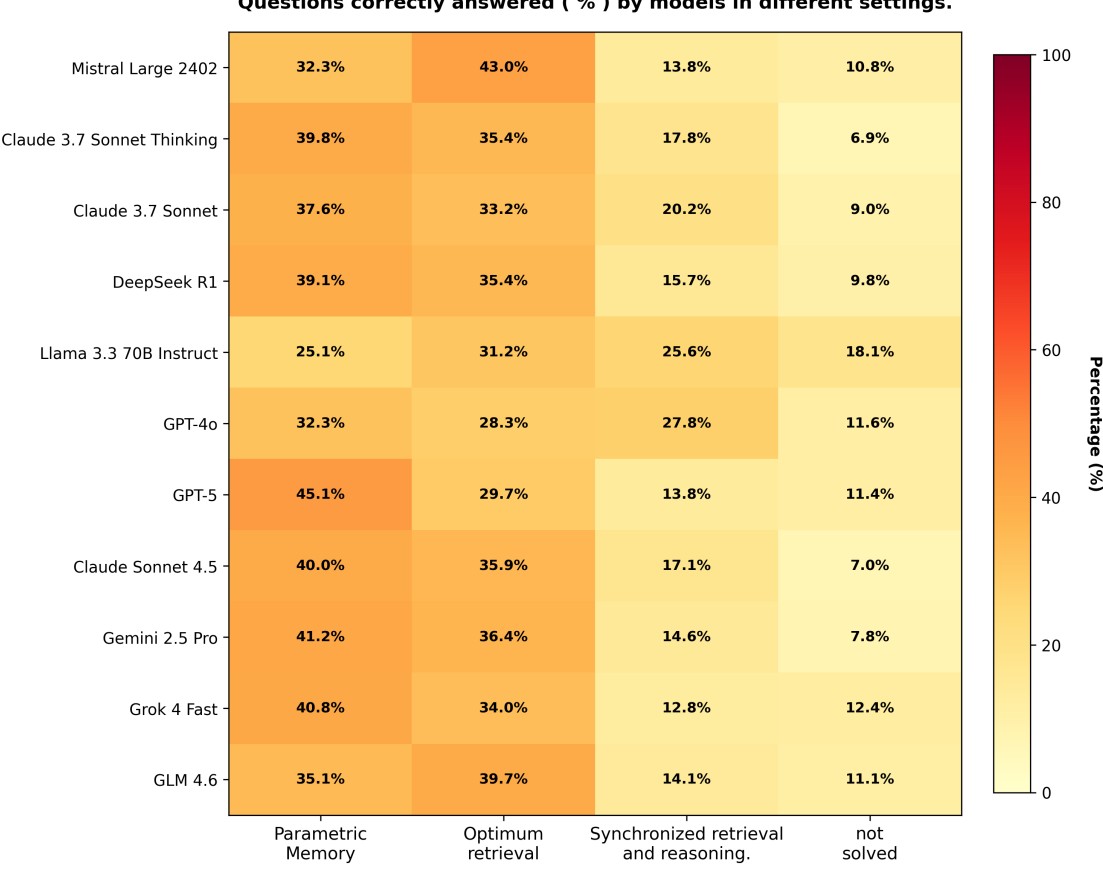

Figure 3: **Partition of Solvability.** This heatmap classifies correct answers by the necessary condition for success: internal knowledge (Parametric), static evidence (Gold-Dependent), or dynamic retrieval (Iterative-Exclusive).

*Suppression*: instances where a model answers correctly using only internal parametric memory (No Context) but answers incorrectly once retrieval is engaged.

Figure 5 quantifies this effect across all evaluated models using the *Parametric Suppression Rate (PSR)*: the percentage of originally correct answers that were suppressed by the retrieval process:

$$PSR = \frac{|Q_{correct}^{NoCtx} \cap Q_{incorrect}^{IterRAG}|}{|Q_{correct}^{NoCtx}|} \tag{1}$$

As illustrated in Figure 5, the average suppression rate across models is 6.7%. However, the variance is significant: Mistral Large 2402 and Llama 3.3 70B show the highest vulnerability, with suppression rates of 14.1% and 11.7% respectively. This suggests these models frequently discard correct internal knowledge in favor of misleading retrieved context. Conversely, the Claude family demonstrates exceptional stability, with Claude 3.7 Sonnet achieving the lowest PSR of 2.7%, followed closely by Claude Sonnet 4.5 at 3.4%. This low rate indicates robust "conflict resolution" capabilities, where the model correctly identifies when to ignore irrelevant retrieved snippets in favor of its parametric memory. Minimizing PSR is a critical safety requirement for enterprise RAG systems to ensure that the addition of retrieval does not degrade the model's baseline competence.

We hypothesize three primary drivers for this behavior. First, Authority Bias, where instruction-tuned models (e.g. Llama3) are optimized to prioritize user-provided context over internal weights, leading them to force alignment with retrieved "distractors." Second, Noise-Induced Uncertainty, where conflicting or

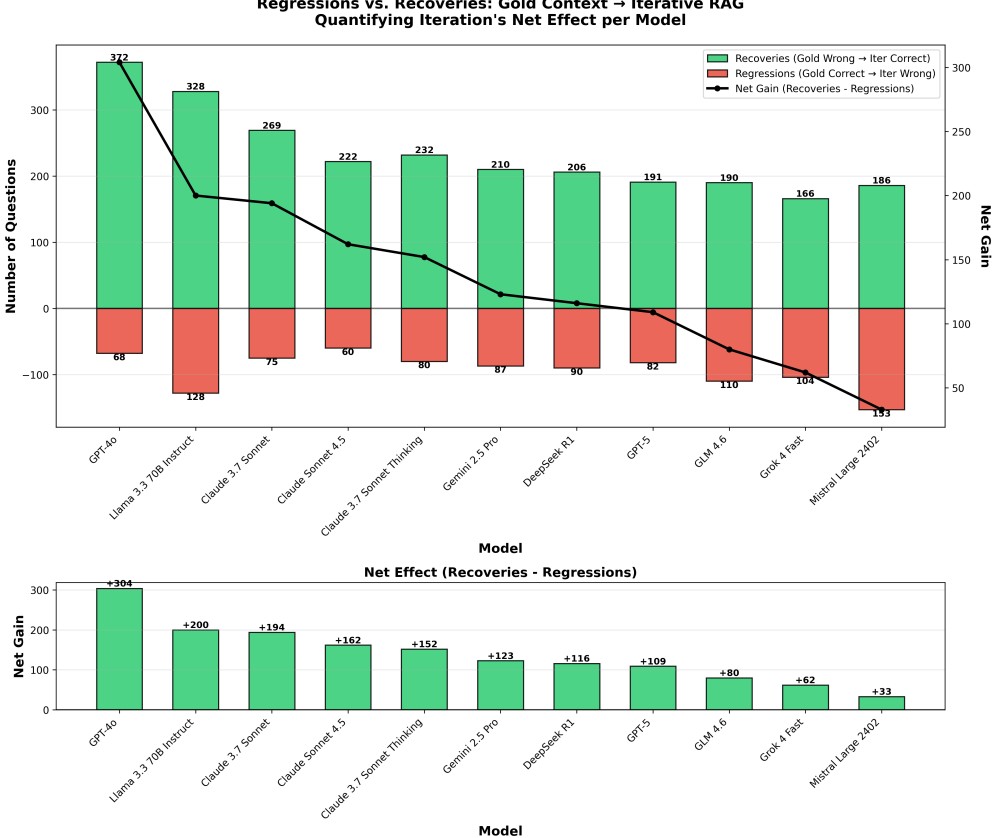

Figure 4: **Recoveries vs. Regressions from Gold Context to Iterative RAG.** Green bars count *recoveries* (Gold incorrect → Iterative correct) and red bars count *regressions* (Gold correct → Iterative incorrect) per model; the black line (top) and the condensed panel (bottom) show the *net gain* questions count (= recoveries − regressions). The plot quantifies iteration's overall benefit: models like GPT–4o and LLAMA 3.3 INSTRUCT post the largest net gains, while MISTRAL LARGE 2402 shows the smallest due to higher regressions.

fragmented evidence increases cognitive load, causing the model to abandon its internal reasoning. Finally, Reasoning Disruption, where an early irrelevant retrieval in a multi-hop chain derails the reasoning path (path drift), causing the model to abandon a correct initial hypothesis.

### 4.5 Portion of Unanswered Questions

Despite the gains offered by Iterative RAG, a stubborn subset of questions remains inaccessible. The share of *not solved* questions varies appreciably across models. CLAUDE SONNET 4.5 and GEMINI 2.5 PRO leave relatively few questions unresolved (about 7–8%), whereas LLAMA 3.3 70B INSTRUCT, GROK 4 FAST, and GPT-4o retain larger unsolved portions (≈11–18%).

Figure 6 aggregates, for each setting, the number of questions that remain *unanswered by all 11 models*, stratified by hop count (1–4). Totals drop sharply from 356 in *No Context* to 73 in *Gold Context* and to 21 in *Iterative RAG*. Contrary to a simple "more hops = harder" assumption, *most* unanswered cases cluster at hop 2 and hop 4, suggesting that difficulty stems from specific bridging steps and long-chain compositions rather than hop count alone. However, Iterative RAG eliminates a large share of these challenging items: unanswered counts fall from $76 \rightarrow 8 \rightarrow 0$ (1 hop), $93 \rightarrow 21 \rightarrow 4$ (2 hops), $85 \rightarrow 15 \rightarrow 4$ (3 hops), and $102 \rightarrow 29 \rightarrow 13$ (4 hops) across No Context, Gold, and Iterative RAG, respectively—evidence that stepwise retrieval with partial-answer state resolves both mid-chain gaps (hop 2) and deeper chains (hop 4).

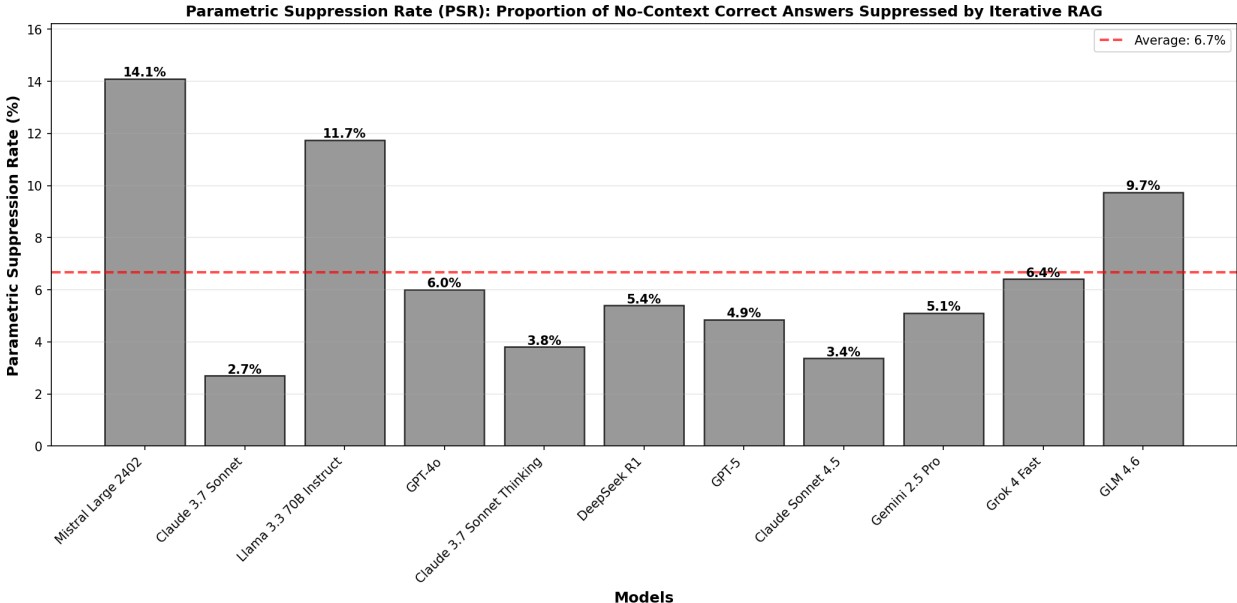

Figure 5: **Parametric Suppression Rate (PSR).** The plot illustrates the proportion of questions answered correctly in the No Context setting that are suppressed (answered incorrectly) in Iterative RAG. Mistral Large 2402 exhibits the highest suppression rate (14.1%), indicating a strong tendency to prioritize retrieved noise over correct internal weights. In contrast, Claude 3.7 Sonnet is highly robust (2.7%), effectively filtering irrelevant context to preserve its parametric knowledge.

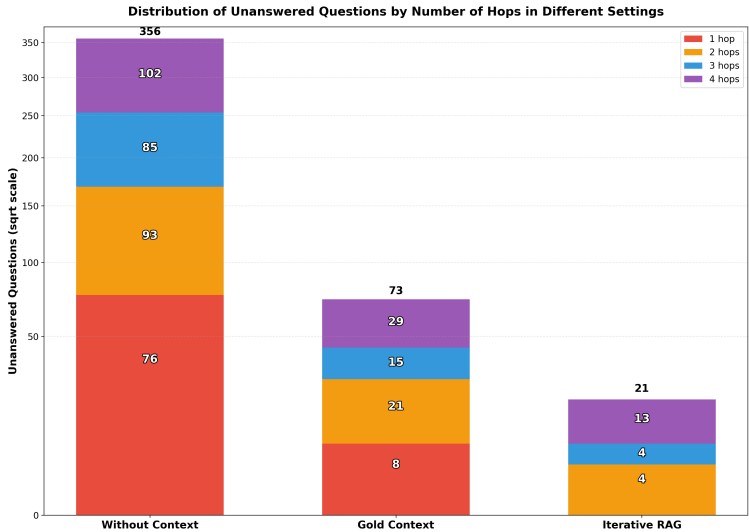

Figure 6: Unanswered questions by all models in different set ups, where model relying on Parametric memory (without context), Ideal static RAG (Gold Context), and synchronized reasoning and retrieval (Iterative RAG).

## 5 Analysis

Our evaluation suggests that large language models benefit significantly when evidence is staged and refined across iterations rather than presented simultaneously. By interleaving retrieval with intermediate reasoning, Iterative RAG enables dynamic query reformulation based on partial hypotheses. This process

lowers cognitive load by restricting the model's focus to a small, high-utility context window at each step, a mechanism that proves beneficial even for reasoning-optimized models. Building on these observations, this section investigates **how** models leverage the Iterative RAG system to achieve these gains and **when** specific failure modes impede their success.

### 5.1 Dynamics of Iterative Utilization

We investigate how different models utilize the iterative pipeline to navigate complex queries, focusing on their ability to self-correct and their efficiency in converging to an answer.

#### 5.1.1 Mechanism of Control: Self-Correction via Anchor Propagation

To understand how iterative control enables self-correction, we analyze the *Anchor Carry-Drop Rate*, defined as the frequency with which a model fails to explicitly propagate a scientific bridge entity (i.e., key chemical entity in this study) from a previous partial answer into the subsequent retrieval query. Since this metric assesses continuity between steps, it is only applicable to runs extending beyond a single retrieval ($t > 1$).

As shown in Figure 7, the data reveals a universal "transition shock" at Step 2, where anchor loss spikes dramatically across all models (e.g., reaching 47.2% for GPT–4o). This high drop rate indicates that models frequently discard their initial hypothesis—constructed solely from the user's original query and the first set of retrieved evidence. Instead of maintaining the initial entities, the models pivot to a different set of anchors, effectively changing the direction of the reasoning path.

Following this re-orientation, the definitive evidence of control is the trajectory observed in Steps 3 through 5. As the reasoning chain deepens, the carry-drop rate consistently declines across all models. This downward slope proves that the model is not searching randomly; after the initial direction change at Step 2, the controller effectively "locks onto" a specific entity chain, maintaining state continuity with increasing precision as it extends the reasoning process. This dynamic behavior allows the system to construct longer reasoning chains and navigate away from initial retrieval sets that do not support further deduction.

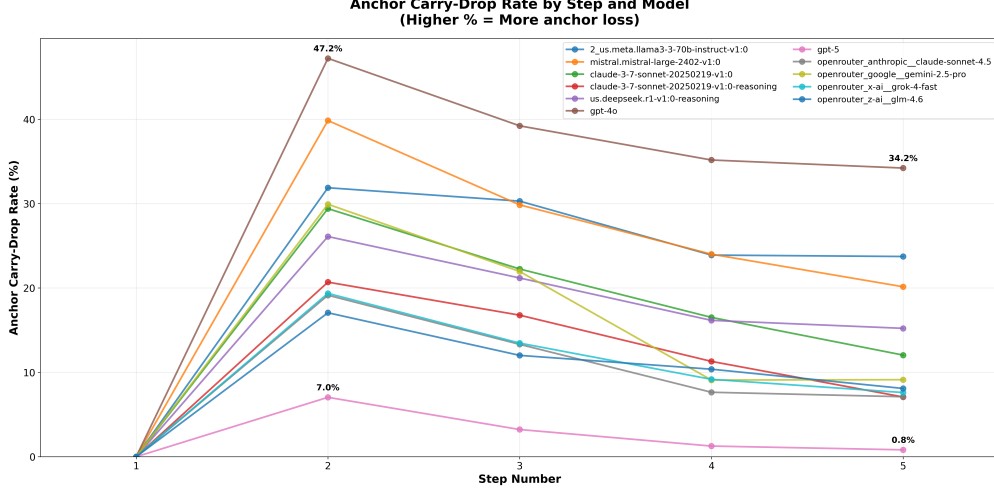

Figure 7: **Anchor Carry-Drop Rate by Step.** A universal spike at Step 2 indicates a "correction pivot," where models discard weak initial hypotheses. The subsequent decline in Steps 3–5 demonstrates reasoning convergence as the controller locks onto the correct entity chain.

#### 5.1.2 Step-Count Distribution and Model Strategy

This active control over the reasoning path results in distinct step-count distributions, revealing how different models strategize their retrieval budget. Figure 8 shows, for each model, performance as a function of the *finalized retrieval step* and the *oracle hop depth* of those questions. The *x*-axis is the step at which the

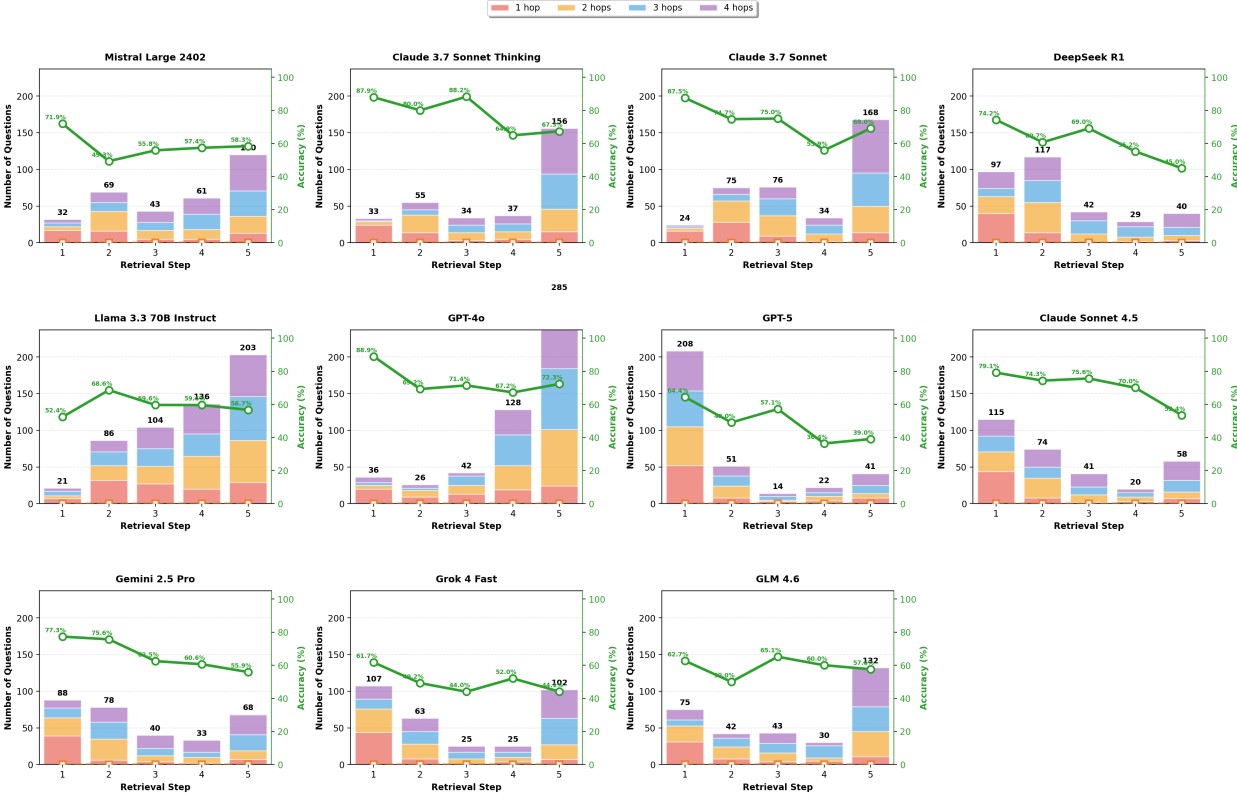

Figure 8: **Performance by finalized retrieval step on questions failed by *Gold-Context* set up.** Stacked bars (right $y$-axis) show the number of questions that finalized at each step, colored by oracle hop depth (1–4). The green solid line (left $y$-axis) reports Iterative-RAG accuracy on exactly those questions.

planner stopped and produced an answer. The green solid line is the *Iterative–RAG accuracy* for the subset of questions that finalized at that step. The stacked bars (right $y$-axis) give the *count of questions* that ended at that step, colored by hop depth (1-4).

**Older models tend to over-iterate**: they often expend *more steps than the task truly requires.* Mistral, Llama, GPT–4o, and Claude 3.7 (both Standard and Thinking) are broadly aligned, with a small but telling difference: Sonnet 3.7s are *somewhat* efficient, using fewer steps for 1-2-hop items and more steps for 3-4-hop items, yet GPT–4o, Mistral and Llama in particular push a large slice of questions to 4-5 steps irrespective of hop depth. This "over–iteration" makes late steps a mixed bag, utilizing more tokens unnecessarily. In Section 5.2, we will define a metric to explain this problem more accurately.

**Newer models flip the story** and target efficiency: they try to answer a substantial share of questions in just 1–2 steps. Still, they diverge in how safely they do it. Claude Sonnet 4.5, the model with strongest performance, maintains relatively high accuracy even when answering in a single step, whereas GPT–5 does not. Another key distinction is how they allocate step 1 across hop depths: GPT–5 addresses a near–uniform mix of 1/2/3/4–hop questions with one step, while Sonnet 4.5 and Gemini 2.5 Pro mostly reserve 1-2 steps for 1-2–hop items and push 3-4–hop chains to later steps. This explains the step–1 accuracy gap ($\sim 64\%$ for GPT–5 vs. $\sim 79\%$ for Sonnet 4.5): GPT–5 is frequently betting on parametric knowledge at step 1 across all hop depths, whereas Sonnet 4.5 fires a step 1 answer primarily when there is a high likelihood of being correct.

A general trend is that accuracy drops as the step index increases. Non–reasoning models typically exhibit a *smaller* late–step drop than reasoning models. For these non–reasoning models, extra steps effectively act as "externalized reasoning," helping them resolve items they initially struggled with; because their

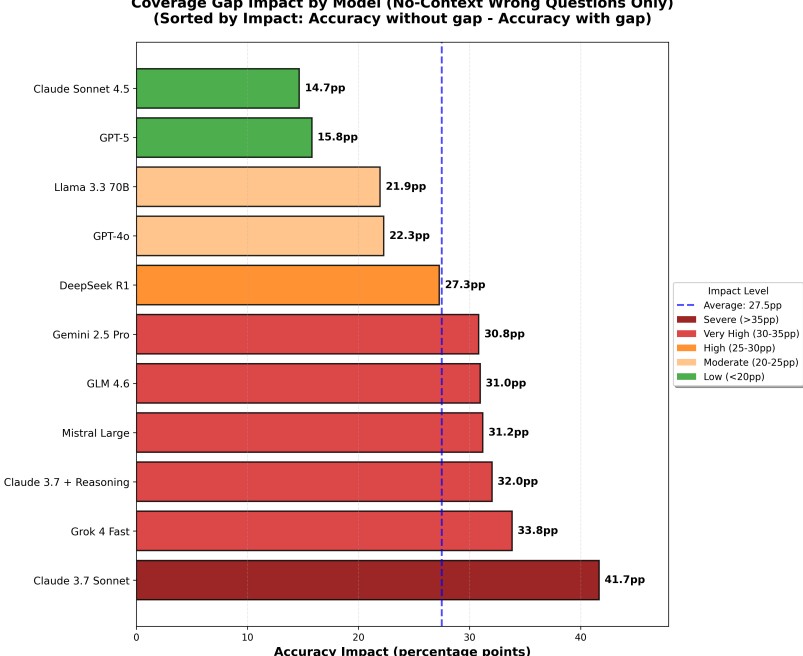

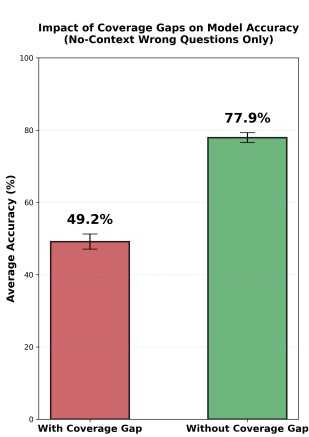

(a) Accuracy with vs. without coverage gaps across models on questions that are not answered in No context. Coverage gaps substantially reduce accuracy ( p value = 0.000001 ). Error bars reflect the standard error of mean

(b) Accuracy with vs. without coverage gaps across models on questions that are not answered in No Context.

Figure 9: Impact of Retrieval Coverage Gap on models.

Gold–Context baseline is weaker, many of the questions they carry into later steps are comparatively easier, so their late–step accuracy holds up better. In contrast, newer reasoning models leave only the *hardest* chains for later steps; thus, when they escalate depth, they encounter a 3-4–hop–heavy pool, and accuracy declines more sharply. Put differently: late steps reflect selection pressure as much as model skill: which question you choose to carry forward determines how fast the green curve falls.

## 5.2 Failure Modes in Iterative RAG

To understand the limitations of an Iterative RAG system, we dissect the specific failure modes that prevent models from reaching the correct answer. We begin by analyzing cases where the introduction of retrieval actively degrades performance. All analysis in this section is limited to the questions that each model had a knowledge gap for and could not answer correctly with its internal parametric memory. This provides a testbed to eliminate the impact of recency of the model updates in this evaluation.

### 5.2.1 Retrieval Coverage Gaps: The Prerequisite for Reasoning

The most consequential failure mode identified is a *Retrieval Coverage Gap*, defined as an instance where at least one oracle hop (a necessary supporting document) is never retrieved at any step of the iterative process.

**Impact on Performance.** To quantify the cost of missing evidence, we measure the accuracy differential between questions with complete retrieval coverage versus those with gaps. We define the *Coverage Gap Impact* ($\mathcal{I}_{gap}$) for a model as the drop in accuracy when a gap is present:

$$\mathcal{I}_{gap} = \text{Acc}(Q_{\text{no-gap}}) - \text{Acc}(Q_{\text{gap}}) \tag{2}$$

Figure 9a quantifies this effect. In aggregate, when *no* coverage gap occurs, average accuracy is 77.9%. However, when a gap is present, accuracy collapses to 49.2%, representing a significant drop and reflecting a severe 28.7 percentage point penalty. This indicates that once the retriever fails to surface a single required

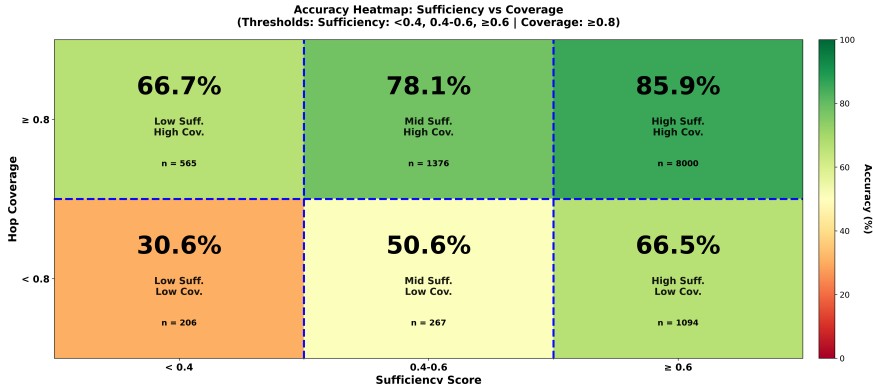

Figure 10: **Sufficiency-Coverage Interaction.** Horizontal axis reflect the sufficiency score and the vertical access reflects the hop coverage in retrieval, and the color illustrate the average accuracy across evaluated models in answering the questions with knowledge gap. The heatmap highlights the "dangerous zone" (low coverage, low sufficiency) where accuracy is lowest (30.6%). Holding sufficiency fixed, improving coverage yields the largest gains (e.g., 30.6% → 66.7%).

hop, most models cannot "reason" their way to the correct answer; success is largely predicated on the prerequisite of covering every hop at least once.

Figure 9b reveals heterogeneous tolerance to these missing hops. Claude Sonnet 4.5 is the most robust, suffering the smallest penalty ($-14.7$ pp), followed by GPT-5 ($-15.8$ pp). The resilience of Llama comes from the overall poor performance of this model. However, for other high resilience models, it likely stems from their strong parametric knowledge. In contrast, most other models incur high to very high penalties ($> 30$ pp), with Claude 3.7 Sonnet exhibiting the highest sensitivity ($-41.7$ pp).

For enterprise RAG applications where answers must be grounded in private corpora (minimizing parametric hallucinations), models that adhere strictly to retrieval coverage like the Claude family are generally preferable, even if they appear more "brittle" to gaps in this specific metric.

### 5.2.2 Evidence Sufficiency vs. Retrieval Coverage

To understand the interplay between finding information (coverage) and effectively using it (sufficiency), we compute two run-level estimators per question:

1. **Sufficiency Score** ($\hat{s} \in [0,1]$): The fraction of sentences in the partial answers that are supported by at least one retrieved snippet.

2. **Hop Coverage** ($\hat{c} \in [0,1]$): The fraction of oracle hops whose key surface entity or relation appears in *any* snippet or partial answer.

Figure 10 bins these metrics into three sufficiency bands ($< 0.4$, $0.4$–$0.6$, $\geq 0.6$) and two coverage regimes ($< 0.8$ vs. $\geq 0.8$), reporting the mean accuracy within each cell. Holding sufficiency fixed, moving from low to high hop coverage produces the largest jumps in accuracy. For low sufficiency, accuracy more than doubles, rising from 30.6% (low coverage) to 66.7% (high coverage). At mid-sufficiency, it increases from 50.6% to 78.1%, and even at high sufficiency, it improves from 66.5% to 85.9%.

In contrast, within a fixed high-coverage regime ($\hat{c} \geq 0.8$), increasing sufficiency yields steadier, incremental gains ($66.7\% \to 78.1\% \to 85.9\%$). A central finding is that *retrieving something about each hop* is the non-negotiable prerequisite; only after coverage is secured do improvements in sentence-level support translate into the final 10–20 percentage points of performance.

The planner's partial answers serve as a communication channel across steps. Models sometimes (i) lean on parametric memory, or (ii) backtrack away from snippets they deem unhelpful. Both behaviors depress

the estimated sufficiency score despite having some hop signals present. This situation could be manageable when coverage is high (green zone in top-left portion of the heat-map in Figure 10, e.g., 66.7% accuracy with high coverage but low sufficiency), but *catastrophic* when coverage is low (30.6%).

### 5.2.3 Confidence Miscalibration: Stopping Too Early or Too Late

The observations of sufficiency score ($\hat{s}$), hop coverage ($\hat{c}$), and the finalized step count lead us to analyze the *controller's* decision-making. We define three confidence states for each run based on the alignment between the model's chosen stopping point and previous diagnostic evidence gathered.

- **Overconfident (Early Finalize):** The model stops before retrieving all required hops (`final_step` < `#hops`) despite having insufficient evidence, defined as $\hat{c} < 0.8$ or $\hat{s} < 0.6$. This represents a premature commitment to an answer without adequate grounding.

- **Underconfident (Over-thinking):** The model continues retrieving even though an earlier step ($t < $ `final_step`) already contained sufficient evidence to support the final answer. This behavior represents inefficient resource usage.

- **Well-Calibrated:** The run is considered well-calibrated if it is neither overconfident nor underconfident, meaning the model stops appropriately when evidence is sufficient or continues the search when it is not.

**Aggregate Impact of Miscalibration.** Figure 11a quantifies the performance cost of these states. *Over-confidence* is the mode of harmful failure: the average accuracy drops to 61.5%, indicating that the models frequently hallucinate or guess when they stop early without enough evidence. In contrast, *Underconfidence* functions mostly as an efficiency tax (by using more steps) rather than a correctness tax, averaging 73.2% accuracy. *Well-calibrated* runs perform best at 74.2%, but not significantly different from the under-confidence setup. Thus, finalizing while both coverage ($\hat{c}$) and sufficiency ($\hat{s}$) and step budgets are low costs a significant drop (roughly 13 percentage points) compared to the other two states.

**Calibration Shifts by Hop Depth.** Figure 11b reveals how model behavior changes as the number of hops increases (from 1 to 4 hops). As hop count increases, several models drift toward overconfidence. DeepSeek R1, GLM 4.6, and Grok 4 Fast show steadily growing "red bands" (overconfidence) by 4-hop questions, with GPT-5 being the most severe. GPT-5 exhibits expected overconfidence particularly in 3-hop and 4-hop scenarios. Conversely, GPT-4o trends Underconfident at 2-3 hops (large blue bands), consistent with a conservative stopping policy despite having adequate coverage. This mirrors the behavior of Claude 3.7 Sonnet (standard and reasoning modes), Llama 3.3, and Mistral Large, which tend to take more steps than necessary, increasing the likelihood of being labeled Underconfident. In contrast, Claude Sonnet 4.5 maintains the most stable profile, keeping overconfidence low with only moderate underconfidence across all hop depths, marking it as the best-calibrated model in our evaluation.

### 5.2.4 Composition Failure: The Synthesis Bottleneck

The next major failure mode we analyze is Composition Failure. This occurs when the retrieval loop successfully places the correct evidence for the final answer into the context window in at least one of the steps, yet the model fails to synthesize it correctly. Unlike coverage gaps where information is missing, this failure mode represents a **squandered opportunity**: regardless of whether the evidence was retrieved through a perfect reasoning chain or a lucky shortcut, the model possessed the raw material for the answer but failed to distinguish the correct signal from the noise.

Specifically, we mark a failure as compositional if, among incorrect answers, the correct entity or claim was present in the retrieved context but the model: (i) selected a different, incorrect entity mentioned in the text (precision drift); (ii) paraphrased the answer vaguely without naming the required entity; or (iii) merged multiple entities, resulting in a factually incorrect or unclear claim.

**Prevalence and Impact:** Figure 12 quantifies the rate of this failure mode. On average, 87.3% of all incorrect answers fall into this category. This is a striking finding: in nearly nine out of ten failures, the

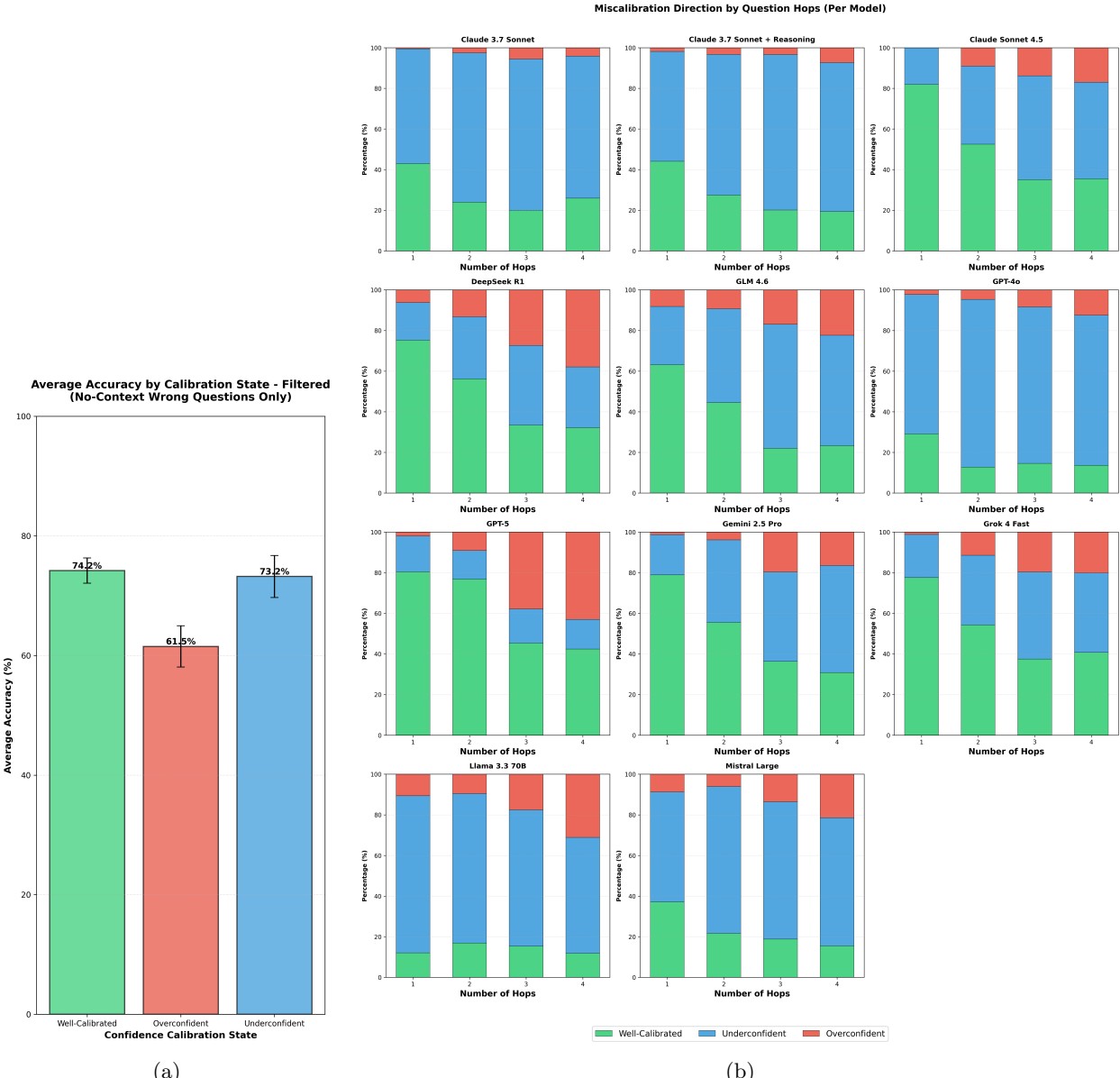

(a)                                                               (b)

Figure 11: **Miscalibration Analysis.** on questions with incorrect answers in No Context setup. a) **Accuracy Impact.** Average accuracy across all models in three states: well-calibrated, under-confident, and over-confident. Overconfidence causes a significant drop in accuracy ($p = 0.0022$ OverConfident vs UnderConfident), while underconfidence primarily affects efficiency. b) **Shift by Hop Depth.** Stacked bar charts showing the percentage distribution of calibration states for questions where the model failed in the No-Context setting, stratified by the number of hops. Most models gradually shift towards overconfidence as the number of hops increases. DeepSeek R1 drifts to overconfidence at 4-hops, while GPT-4o remains predominantly underconfident.

system is not limited by the reach of its retriever, but by the fragility of its reasoning. The model essentially "stumbles at the finish line," unable to faithfully extract the answer it has already found.

To validate that this is a reasoning deficit rather than a retrieval artifact, we further restricted our analysis to "perfect retrieval" scenarios, cases where the model successfully retrieved *every single oracle hop* required for the reasoning chain (see Appendix Figure S7). Even under these ideal conditions, where the entire logical

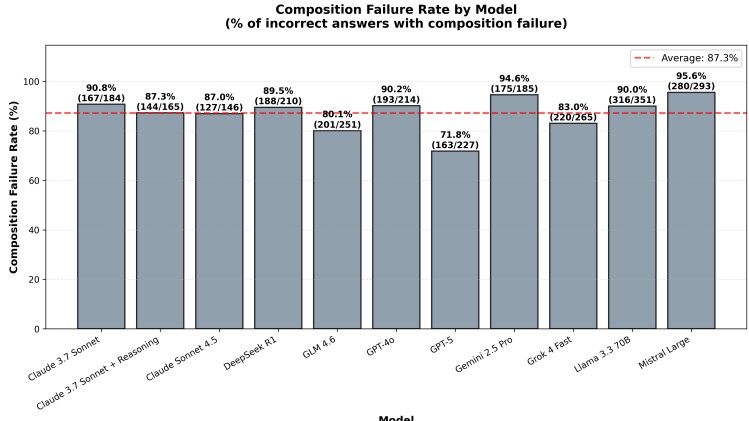

Figure 12: Composition Failure Rates. The percentage of incorrect answers where the correct evidence was retrieved but not used correctly. High rates indicate a limitation in answer synthesis.

path is visible in the context, the average composition failure rate remains at 58.6% across all errors. This confirms that for the majority of failures, the bottleneck is indeed the synthesis of complex information, not the loss of intermediate reasoning steps.

The rate varies by model architecture. The highest composition failure rates are observed in Mistral Large (95.6%, 280/293) and Gemini 2.5 Pro (94.6%, 175/185). For these systems, the primary bottleneck is not finding information, but processing it correctly at the final step. Mid-tier rates are represented by models such as GPT-4o (90.2%, 193/214) and DeepSeek R1 (89.5%, 188/210). The lowest rates are found in GPT-5 (71.8%, 163/227) and GLM 4.6 (80.1%, 201/251). This suggests that while some models like GPT-5 are prone to missing evidence entirely (coverage gaps), models like Mistral are highly effective at finding documents but lack the discrimination capability to utilize them under distraction.

### 5.2.5 Distractor Latch: The Retrieval Trap

As reported in Figure 8, we observed a counter-intuitive trend where questions requiring more retrieval steps often yielded lower accuracy. While one might attribute this simply to the increased difficulty of such questions, a critical structural risk emerges with longer retrieval chains: the accumulation of context increases the probability of encountering distractors.

We define *Distractor Latch* as a run-level failure where the retrieval loop "locks onto" a chemically similar but *irrelevant* scaffold or family (e.g., retrieving "benzylic" entities when the oracle path requires "phenoxyl") and reinforces that track in subsequent steps. Concretely, we match family/entity patterns in snippet text against the oracle hop entities; if the dominant retrieved family is off-target, we flag the run as a latch.

**Impact and Prevalence.** Distractor Latch is devastating. As shown in Figure 13a, the average accuracy collapses from 84.8% (No Latch) to 30.9% (Has Latch), a massive penalty of 53.9 percentage points. The error bars in Figure 13a reflect that this degradation is consistent across all evaluated models; even strong controllers drop significantly under distractor latch conditions.

While this phenomenon is universal, the frequency varies significantly by model (Figure 13b). Mistral Large (24.2%) and Llama 3.3 70B (21.7%) latch most often to incorrect entities, indicating weaker discrimination between relevant and irrelevant chemical scaffolds. They are followed by GPT-4o (19.1%) and DeepSeek R1 (18.6%). GPT-5 latches the least frequently (11.1%), showcasing superior understanding of texts and questions.

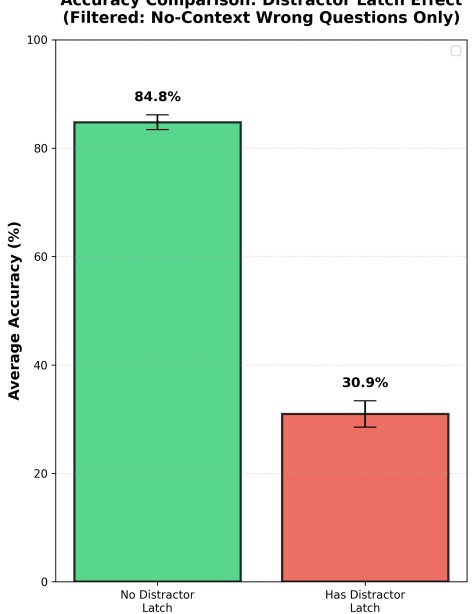

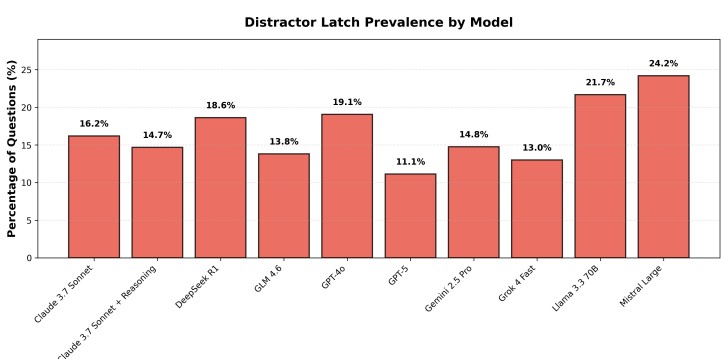

(a) **Impact on accuracy.** Accuracy drops from 84.8% to 30.9% when a distractor latch occurs.

(b) **Prevalence.** Fraction of questions affected by distractor latch for each model.

Figure 13: **Analysis of distractor latch effects.** Panel (a) shows the effect of distractor latch on accuracy, computed only over questions that were answered incorrectly in the No Context setup. Panel (b) reports how frequently each model falls into this failure mode. The presence of a distractor latch imposes a large accuracy penalty of approximately 53.9 percentage points, and all evaluated models are affected to some degree.

## 5.3 Efficiency Analysis

Beyond raw accuracy, the viability of iterative RAG systems in production environments is governed by their computational efficiency and economic scalability. In this section, we analyze the trade-offs between performance and resource consumption, explicitly mapping the **cost-accuracy** and dissecting the tension between **adaptive reasoning effort** and **token consumption predictability**. Our analysis reveals distinct model archetypes, from high-value specialists to volatile reasoners, that necessitate different deployment strategies depending on the application's tolerance for cost and variance.

Figure 14 illustrates the relationship between the total cost (calculated based on input/output tokens and retrieval calls and the cost of API calls) and the final accuracy on the benchmark. The X-axis represents the total cost in USD for running the full evaluation set, while the Y-axis denotes the accuracy percentage.

The distribution reveals a non-linear relationship where diminishing returns set in at higher cost brackets. We observe three distinct behaviors:

- **High-Value Specialists:** Models like Grok 4 Fast, GLM 4.6, and DeepSeek R1 occupy the "value corner" (mid-left), delivering strong performance for minimal investment. DeepSeek R1, in particular, achieves an accuracy ($\approx 82.5\%$) comparable to GPT-4o and GPT-5 but at significantly lower cost ($\approx \$40$ vs. $\$60 - \$80$). This suggests that specialized reasoning distillation can offer a superior ROI for multi-hop tasks compared to general-purpose scaling.

- **The Premium Frontier:** The Claude family (Sonnet 3.7, 4.5) and Gemini 2.5 Pro push the accuracy ceiling ($\geq 84\%$). Notably, Claude Sonnet 4.5 appears to be the most capable model,

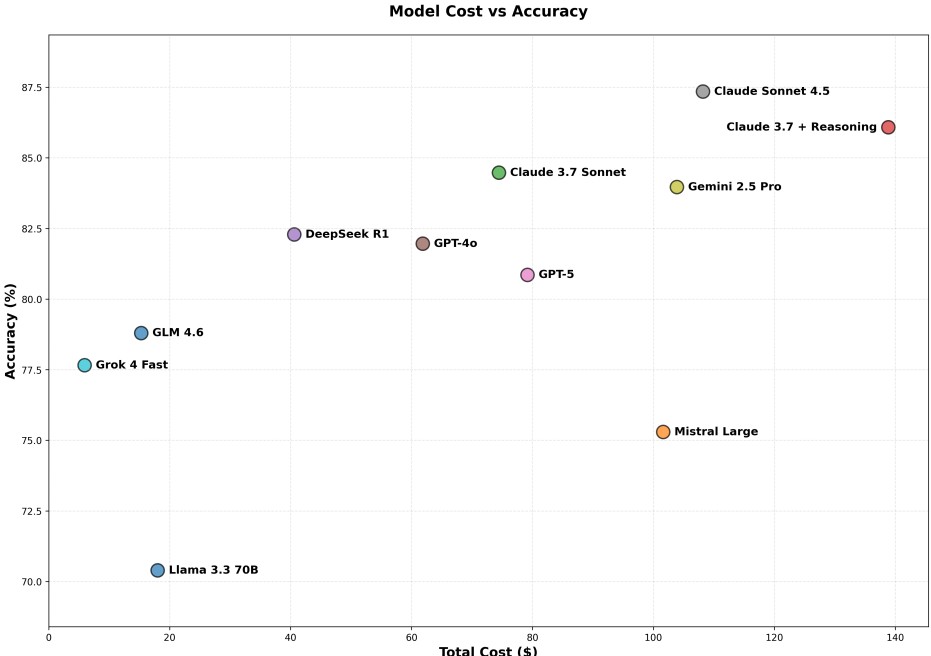

Figure 14: Iterative RAG Cost vs Accuracy per model.

surpassing Claude 3.7 + Reasoning in accuracy while incurring lower costs ($\approx$ \$110 vs. \$140), indicating algorithmic improvements in efficiency over the brute-force "extended thinking" approach.

- **Inefficiency Traps:** Mistral Large stands out as an outlier in inefficiency, with costs exceeding \$100 (comparable to Gemini/Claude) but accuracy lingering near 75%. This highlights that parameter count or generation pricing does not strictly correlate with reasoning capability in iterative settings.

### 5.3.1 The Trade-off: Adaptivity vs. Predictability

A central challenge in deploying reasoning-heavy models is managing the tension between the ability to "think longer" on complex problems (Adaptivity) and the requirement for stable system latency (Predictability). Figure 15 visualizes this trade-off by mapping models along two dimensions:

- **Token Scaling Factor ($S$):** This metric measures how much a model increases its computational effort when faced with harder problems. It is calculated as the ratio of the average token count for hard questions ($Q_{hard}$, where 9-11 models failed) to easy questions ($Q_{easy}$, where 0-2 models failed):

$$S = \frac{\mu_{tokens}(Q_{hard})}{\mu_{tokens}(Q_{easy})} \qquad (3)$$

A value of 2.0$\times$ implies the model doubles its generation length for difficult queries.

- **Token Usage Consistency ($C$):** This metric quantifies the "noise" or volatility in output length within similar difficulty levels. It is defined as the mean Coefficient of Variation (CV) across difficulty buckets ($D = \{easy, medium, hard\}$):

$$CV = \frac{1}{3} \sum_{d \in D} \left( \frac{\sigma_{tokens}(Q_d)}{\mu_{tokens}(Q_d)} \times 100 \right) \qquad (4)$$

Lower values indicate highly predictable latency profiles, while higher values signal erratic generation lengths.

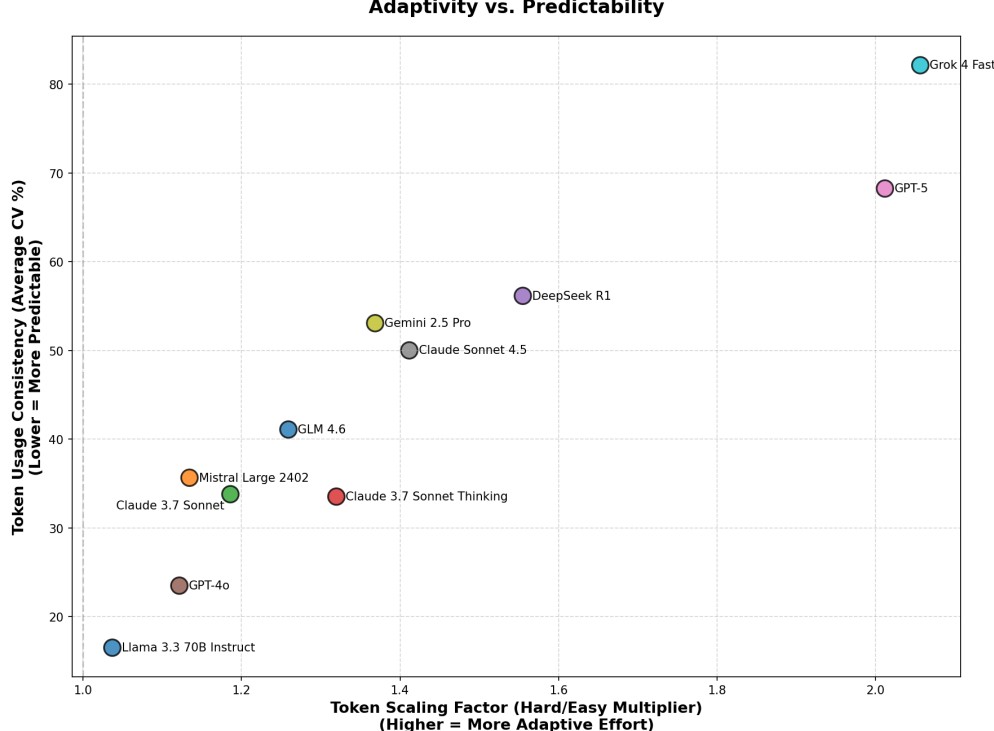

Figure 15: **The Trade-off between Adaptivity and Predictability.** Models are plotted by their *Token Scaling Factor* (X-axis) against their *Token Usage Consistency* (Y-axis). A clear diagonal trend emerges: aggressive adapters (e.g., Grok 4 Fast, GPT-5) incur a high "volatility tax" ($CV > 65\%$). Conversely, "Rigid Executors" (e.g., Llama 3.3, GPT-4o) remain highly predictable ($CV < 30\%$) but lack dynamic scaling. Claude Sonnet 4.5 strikes a balance, offering moderate adaptivity with manageable variance.

This combined view reveals three distinct behavioral archetypes:

- **The Rigid Executors (Bottom-Left):** Models such as Llama 3.3 70B and GPT-4o cluster in the extreme bottom-left. They exhibit minimal scaling ($< 1.2\times$), applying nearly identical cognitive effort regardless of whether a question is easy or hard. While this limits their reasoning depth on complex tasks, it yields exceptional operational consistency ($CV < 30\%$). This profile offers the most predictable latency for Service Level Agreement (SLA)-sensitive applications, albeit at the cost of reasoning flexibility. Notably, Claude 3.7 Sonnet (Thinking) maintains this high consistency ($CV \approx 33\%$) even while offering slightly higher scaling ($1.3\times$) than the pure rigid executors.

- **The Volatile Reasoners (Top-Right):** In sharp contrast, Grok 4 Fast and GPT-5 occupy the top-right extreme. These models display aggressive adaptivity, generating over $2.0\times$ more tokens for hard questions than easy ones. However, Figure 15 highlights the "volatility tax" of this behavior: their output variance surges to $70 - 80\%$ CV. This extreme unpredictability makes cost forecasting and latency estimation difficult, as the model's compute load fluctuates wildly based on the specific query complexity.

- **The Balanced Strategists (Center):** Claude Sonnet 4.5 and Gemini 2.5 Pro strike a middle ground. With scaling factors around $1.4\times - 1.5\times$, they demonstrate the capacity to expand reasoning steps when necessary without succumbing to the extreme variance of the pure reasoning models. This "sweet spot" ($CV \approx 50\%$) suggests a more controlled implementation of adaptive computation, where additional tokens are allocated efficiently rather than expansively.

Ultimately, the choice of model depends on the application's tolerance for variance: while the *Volatile Reasoners* provide a higher reasoning ceiling for complex hops, the *Rigid Executors* offer the stability required for high-volume production environments. Appendix S1.7 further shows that Iterative RAG improvements are not explained solely by Gold Context length, since improved questions are not concentrated only in the longest-context bins.

### 5.4 Adherence and Controllability

Beyond accuracy and failure rates, the utility of an Iterative RAG system depends on its *controllability*. A critical dimension of control is the model's adherence to system instructions—specifically, the instruction to utilize the iterative retrieval pipeline for verification, even when it believes it possesses sufficient parametric knowledge to answer the question directly. We term this behavior *Procedural Compliance*. In high-stakes domains, a model that skips verification due to confidence in its internal memory poses a safety risk. Therefore, we seek models that respect the *process* of evidence acquisition as strictly as the *outcome* of the answer.

**Procedural Compliance** To measure procedural compliance, we isolate the subset of questions $Q_{known}$ that each model answered correctly in the *No Context* setting. These are questions where the model ostensibly does not *need* external help. We further restrict this pool to questions with *more than one hop* ($N_{hops} > 1$) to specifically target multi-hop reasoning. Additionally, since our implementation enforces a mandatory first retrieval step, its presence in single-hop questions does not reflect the controller's active adherence to retrieval instructions.

The **Procedural Compliance Rate (PCR)** is defined as the proportion of known questions where the model either successfully verified the answer or actively attempted to do so:

$$PCR = \frac{|Q_{known} \cap (\text{Effective} \cup \text{Ineffective})|}{|Q_{known}|} \tag{5}$$

The models' behavior on this filtered $Q_{known}$ set falls into three distinct categories:

(i) **Effective Compliance:** The model actively uses the tool (taking at least 2 steps) and successfully retrieves evidence covering the full reasoning path (No Coverage Gap). This is the ideal behavior: the model knows the answer and successfully backs it up.

(ii) **Ineffective Compliance:** The model demonstrates effort by continuing the search beyond the mandatory minimum ($Steps > 1$) but fails to find complete evidence (Coverage Gap). This indicates adherence to the protocol (trying to fill the gap) despite retrieval limitations.

(iii) **Non-Compliance:** The model stops at the mandatory minimum ($Steps = 1$) despite having a coverage gap (Coverage Gap). This suggests the model "gave up" on verification immediately and relied solely on its parametric memory, prioritizing efficiency over the instructed evidence-gathering process.

Table 3 presents the results. We observe high adherence across most models, with Claude 3.7 Sonnet achieving perfect compliance (100%), it never showed Non-Compliance on multi-hop questions it knew. Llama 3.3 and Mistral Large also show exceptional adherence (> 97%). However, GPT-5 emerges as a significant outlier, exhibiting frequent Non-Compliance. It bypassed the verification protocol in 31.8% of cases (124 runs), resulting in a PCR of only 68.2%. This suggests a trade-off in newer frontier models: while capable, they may be harder to constrain to strict evidence-gathering protocols compared to models like Claude or Llama.

**Success Rate of Compliance Attempts** Beyond the willingness to comply, we also assess the *effectiveness* of these attempts by calculating the Success Rate: the proportion of compliant runs that were actually "Effective." Based on our observations, Claude 3.7 Sonnet is not only perfectly compliant but also highly effective, successfully verifying its knowledge in 90.5% of its compliant runs (276/305). Similarly, Claude Sonnet 4.5 and Gemini 2.5 Pro demonstrate robust retrieval capabilities, converting their compliance into

Table 3: **Procedural Compliance Rates (PCR).** The table analyzes behavior on multi-hop questions correctly answered in No-Context. **Effective** indicates successful retrieval of the full chain; **Ineffective** indicates active search ($> 1$ step) that missed evidence; **Non-Compliant** indicates skipping verification (stopping at step 1 with gaps). **Success Rate** measures the effectiveness of compliant runs ($\frac{\text{Effective}}{\text{Effective+Ineffective}}$).

| Model | $|Q_{known}|$ | Effective | Ineffective | Non-Compliant | PCR (%) | Success Rate (%) |
|---|---|---|---|---|---|---|
| Claude 3.7 Sonnet | 305 | 276 | 29 | 0 | **100.00** | 90.5 |
| Claude 3.7 + Reasoning | 325 | 284 | 35 | 6 | 98.15 | 89.0 |
| Llama 3.3 70B | 207 | 135 | 68 | 4 | 98.07 | 66.5 |
| Mistral Large | 279 | 224 | 47 | 8 | 97.13 | 82.7 |
| GPT-4o | 263 | 218 | 34 | 11 | 95.82 | 86.5 |
| Claude Sonnet 4.5 | 337 | 298 | 23 | 16 | 95.25 | 92.8 |
| Gemini 2.5 Pro | 352 | 297 | 35 | 20 | 94.32 | 89.5 |
| DeepSeek R1 | 330 | 265 | 45 | 20 | 93.94 | 85.5 |
| GLM 4.6 | 289 | 232 | 32 | 25 | 91.35 | 87.9 |
| Grok 4 Fast | 254 | 204 | 27 | 23 | 90.94 | 88.3 |
| GPT-5 | 390 | 237 | 29 | 124 | 68.21 | **89.1** |

verified answers with success rates of 92.8% and 89.5%, respectively. In contrast, Llama 3.3 70B, while highly compliant (98.1% PCR), struggles to actually find the evidence, verifying only 66.5% of its attempts. This indicates that while Llama is "obedient" and tries to use the tool, its retrieval planning capabilities are weaker than the Claude or Gemini families, leading to many "Ineffective Compliance" outcomes. GPT-5, despite its high rate of Non-Compliance, has a high success rate when it *does* choose to comply (89.1%), suggesting its failure is purely behavioral (refusal to use the tool) rather than capability-based.

# 6 Discussion

Our study finds that LLMs answer scientific multi-hop questions more reliably when evidence is staged and refined across iterations rather than presented simultaneously. Interleaving retrieval with reasoning enables dynamic query reformulation and keeps the model's focus on a small, high-utility context window, improving accuracy even for reasoning-optimized models. Similar gains appear in other iterative RAG systems: Iter-RetGen and CoRAG alternate retrieval and reasoning to expose missing knowledge, outperforming strong single-shot baselines (Shao et al., 2023; Park et al., 2025c). Structured approaches like KiRAG and Know-Trace confirm this pattern in knowledge-graph setups, showing that incremental query rewriting and context filtering mitigate overload while boosting multi-step reasoning (Fang et al., 2025; Li et al., 2025a). Together, these results highlight a shared principle: staged retrieval beats bulk "ideal evidence" by reducing distraction and supporting adaptive reasoning.

Our results suggest that performance on complex scientific multi-hop questions is governed by three interacting factors: (i) the model's reasoning capacity, (ii) the quality of retrieved evidence, and (iii) the synchronization between retrieval and reasoning. The effect of reasoning independent of retrieval is most directly reflected in the No Context setting. Models with explicit reasoning budgets or reasoning-oriented tuning generally perform better than models that answer directly from parametric memory. A particularly controlled comparison is Claude 3.7 Sonnet in its Standard and Thinking configurations: both share the same model family and training background, but the Thinking configuration has an explicit reasoning budget, improving No Context accuracy from 37.52% to 39.80%.

The effect of retrieval is isolated by the Gold Context condition. Here, we simulate an idealized static retrieval setting by providing the oracle supporting paragraphs from which the multi-hop question was generated. This removes retrieval noise and avoids introducing unrelated distractor documents, although in a multi-hop setting one oracle paragraph may still be distracting when the model is answering a different hop-level sub-question. Both reasoning-oriented and non-reasoning models benefit substantially from this idealized evidence condition: average accuracy increases from 37.16% in No Context to 69.14% in Gold Context, and reasoning models often reduce their output length because they no longer need to compensate for missing evidence through extended internal reasoning.

Finally, the Iterative RAG setting evaluates the joint optimization of retrieval and reasoning. The Gold+CoT ablation provides an intermediate diagnostic baseline: it combines static oracle evidence with an explicit multi-step reasoning prompt, resembling part of the reasoning structure encouraged by Iterative RAG while keeping evidence presentation fixed. This improves performance for both reasoning and non-reasoning models, showing that structured reasoning over retrieved evidence is important, but it does so at the cost of higher computation and still remains below Iterative RAG. Moreover, Gold Context and Gold+CoT are diagnostic upper-bound settings rather than practical single-shot retrieval systems: in real multi-document multi-hop QA, a single static retrieval step may not reliably retrieve all oracle paragraphs needed across different hops. Iterative RAG provides a more operational mechanism by repeatedly retrieving targeted evidence, updating the partial answer state, and aligning the next query with the model's current reasoning trajectory. The additional gains over both Gold Context and Gold+CoT therefore indicate that the key advantage is not retrieval alone or reasoning alone, but the staged synchronization of the two.

This observation can be backed by neuroscience findings. Cognitive Load Theory (CLT) and working-memory research show that performance improves when extraneous load is minimized and attention is focused on a few chunks, conditions met by iterative RAG's narrow windows (Sweller, 1988). Zhang (2025) suggests cognitive-load-aware inference, formalizing cognitive load theory for LLMs, and demonstrates that structured inference reduces token usage and improves reasoning, while benchmarks confirm that context saturation degrades multi-hop performance. EEG studies further reveal that interaction-aware, staged reasoning aligns with neural markers of attention and effort. Thus, our finding that iterative RAG beats ideal evidence is reinforced by both computer science and neuroscience: reducing extraneous load through staged retrieval-reasoning loops aligns with human cognitive limits and yields more robust machine reasoning.

## 7 Conclusion

The results of this work demonstrate that the structure and timing of retrieval can have a greater impact on multi-hop reasoning performance than the completeness of the evidence itself in scientific domain. Across all evaluated models, the precision increases dramatically from $37.16 \pm 5.54\%$ in the No-context condition to $69.14 \pm 7.22\%$ with the oracle evidence, yet Iterative RAG achieves $80.89 \pm 5.8\%$, with the best model surpassing the best oracle-evidence system by 13.8 percentage points. This pattern indicates that retrieval is not merely a mechanism for supplying missing information; rather, its interaction with a model's evolving intermediate reasoning plays a decisive role in determining final performance. The particularly large gains for non-reasoning models suggest that iterative retrieval provides a form of external cognitive scaffolding that compensates for the absence of internal chain-of-thought mechanisms.

These findings highlight a broader conceptual insight: the effectiveness of multi-hop QA systems emerges not from evidence quality alone, but from alignment between the retrieval trajectory and the model's reasoning trajectory. Even with near-perfect document coverage, failures concentrated around composition, distractor latch, and stability across hops reveal that synthesis remains the dominant bottleneck in scientific reasoning. The observed interplay between retrieval order, attention allocation, and inference stability explains why static oracle evidence is insufficient and why dynamically structured retrieval enables qualitatively different reasoning behavior.

In this work we demonstrated iterative retrieval can outperform idealized evidence conditions and provided a principled, model-agnostic diagnostic framework that exposes previously hidden mechanisms underlying multi-step failures. By evaluating a broad set of architectures under controlled orchestration, this study uncovers characteristic behavioral signatures—such as parametric-memory suppression, anchor-drift across hops, and evidence-integration instability—that cannot be identified from aggregate accuracy alone. This mechanistic understanding provides a foundation for designing retrieval controllers and reasoning pipelines that are explicitly adapted to model-specific strengths and vulnerabilities.

Future work can build on these insights by developing adaptive controllers that dynamically allocate retrieval steps based on model uncertainty; by training retrieval and reasoning components jointly to mitigate composition drift; and by incorporating structured memory or graph-based representations to reduce early-hop instability. Extending the diagnostic framework to other scientific domains may further clarify the generality of the mechanisms identified here. Ultimately, the evidence suggests that progress in multi-hop scientific QA

will come from systems that treat retrieval as an active and co-evolving part of the reasoning loop rather than as a static source of evidence, pointing toward a new generation of retrieval-aligned, reliability-oriented model architectures.

## Acknowledgments

The author(s) gratefully acknowledge the financial support for the research, authorship, and/or publication of this article provided by *MITACS* under funding number IT32409. We also thank *Jürgen Müller* for his leadership and project management expertise; *Tobias Roth* for his essential contributions to infrastructure; *Mohammad Khodadad* for his invaluable recommendations. We used AI-assisted tools for proofreading and improving the clarity of the manuscript.

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

## S1 Appendix

### S1.1 Extended Literature Review

The 2024–2025 period brought a steady run of dense-transformer LLMs, with some Mixture-of-Experts (MoE) experiments. In February 2024, Mistral released *Mistral Large*, a conventional dense transformer tuned with instruction Supervised Fine-tuning (SFT) for predictable, long-context use AI (2024b). Its test-time scaling is intentionally modest: a few examples, slightly longer prompts, and quick self-checks help, but gains level off once the prompt already supplies good evidence. In May 2024, OpenAI introduced *GPT-4o*, continuing the dense-transformer line with instruction SFT and preference tuning; extra tokens at inference (e.g., worked examples or short verification) usually yield steady, incremental improvements OpenAI (2024a). Anthropic's *Claude 3.7 Sonnet* (2024–2025) is also dense, offered in configurations that either stay compact and alignment-first or allow more deliberate "extended thinking"; both are SFT-first, with the latter designed to convert added inference budget into more structured intermediate steps when the prompt provides solid evidence Anthropic (2025a).

By December 2024, Meta's *Llama 3.3 70B Instruct* provided an open, instruction-tuned dense baseline AI (2024a). Its size (70B) is practical for self-hosting; test-time gains mostly come from clean gold context, few-shot prompts, and light self-reflection rather than long chains. In January 2025, *DeepSeek-R1* appeared with a staged recipe: SFT on reasoning traces, then preference/RL to stabilise multi-step inference; when you spend extra tokens at test time, it is more willing to run short hypothesis-and-check loops that stick (assuming retrieval is cooperative) Guo et al. (2025c). Around the same time, Google's *Gemini* line kept iterating; official notes describe dense backbones, SFT-first tuning, and regular 2025 updates. The scaling story is consistent: more context, a few exemplars, and brief verification passes help without encouraging needless verbosity Google (2025). Reasoning-oriented training (e.g., DeepSeek-R1's RL-only schedule) provides useful test-time structure for multi-step RAG, helping convert extra tokens into deliberate intermediate reasoning (Guo et al., 2025b;a).

Mid to late 2025, the "frontier" refresh arrived. *GPT-5* (Aug 2025) stays dense and SFT+preference tuned; the goal is smoother, less jumpy gains when users add exemplars, reasoning tokens, or clean retrieval OpenAI (2025). Anthropic's *Claude 4.5 Sonnet* (Sept 2025) continues the dense line with stronger longer-horizon control; its SFT variants try to ensure that spending more tokens yields helpful intermediate structure rather than drift Anthropic (2025b). xAI's *Grok* evolved from an MoE phase (e.g., the 314B Grok-1) toward later, reasoning-oriented post-training on streamlined backbones; instruction SFT and preference-style objectives enable modest few-shot and self-check gains without long, fragile chains Wikipedia contributors (2025); xAI (2025). Finally, ZhipuAI's *GLM 4.6* (late Sept 2025) extends a pragmatic dense family for enterprise coding/reasoning; instruction SFT supports the usual test-time levers (few-shot conditioning, short intermediate reasoning, and measured benefits from tidy retrieval).

Overall, three overlapping currents are relevant to this work: (i) *Alignment-first dense transformers* (e.g., Mistral Large 2402; Llama 3.3 70B) favour controllability and predictable serving, where SFT gives steady but conservative test-time gains; (ii) *Reasoning-oriented variants* (e.g., Claude 3.7 "extended thinking"; DeepSeek-R1's staged SFT+preference/RL) are calibrated to turn extra tokens into short, useful intermediate steps; and (iii) *Frontier dense generations* (GPT-4o → GPT-5; Gemini 2025; Claude 4.5) aim for calm long-context behaviour so that added context, exemplars, and light verification pay off without bloating the response.

## S1.2 Performance and Token Utilization

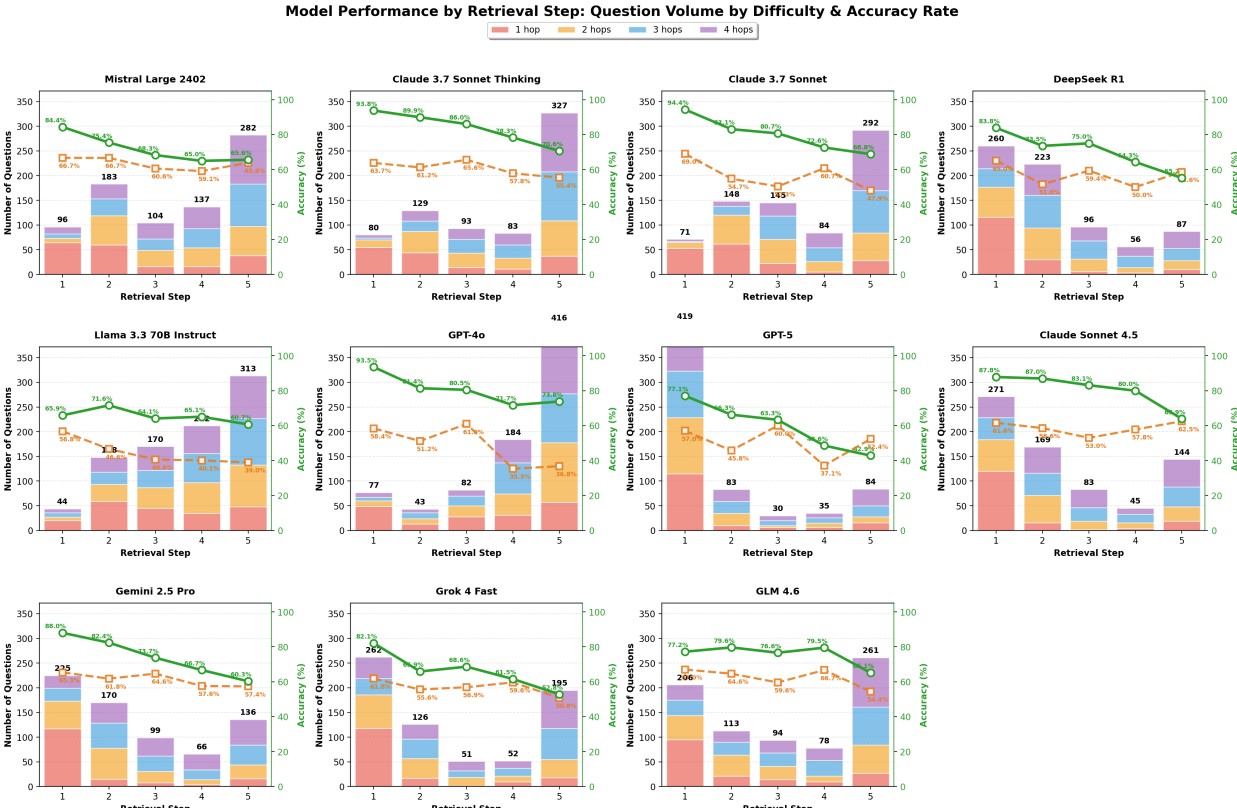

Figure S1: **Model Performance by Retrieval Step (No-Context wrong).** Bars show the number of questions handled at each step, stacked by hop depth (1–4). Lines show accuracy for Iterative RAG (green, solid) and Gold Context (orange, dashed). Early steps capture most 1–2-hop repairs; later steps concentrate 3–4-hop leftovers where accuracy declines. Front-loaded controllers (e.g., Sonnet 4.5, Gemini 2.5 Pro, Claude 3.7) gain early but are drop-sensitive if they over-iterate, while depth-tolerant controllers (e.g., GLM 4.6, Llama 3.3) peak later with smaller late-step losses.

## S1.3 Detailed Failure Mode Analysis

To provide a granular view of model reliability, we decompose failure modes into three components: frequency, severity, and total expected damage.

### S1.3.1 Prevalence and Impact

First, we analyze **Prevalence** ($p_{m,f}$), which measures how often a specific failure occurs. Table S1 highlights the frequency of Retrieval Coverage Gaps, while Table S2 details the prevalence of Overconfidence

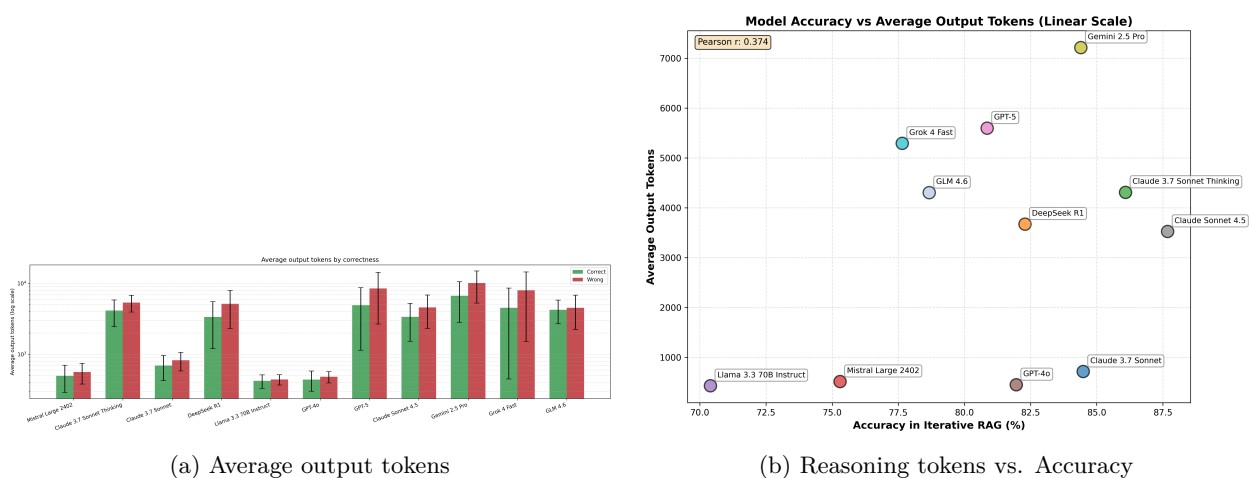

(a) Average output tokens         (b) Reasoning tokens vs. Accuracy

Figure S2: Token utilization patterns across models, revealing three distinct reasoning strategies.

and Distractor Latching. Second, we evaluate **Impact** ($\Delta_{m,f}$), quantifying the penalty—specifically, the drop in accuracy (in percentage points) when a failure is present compared to when it is absent (Table S3).

Table S1: Retrieval Coverage gap prevalence by model (%).

| Model | Coverage Gap Rate (%) |
|---|---|
| GPT-5 | 28.95 |
| Llama 3.3 70B | 26.73 |
| DeepSeek R1 | 16.86 |
| GLM 4.6 | 15.79 |
| Grok 4 Fast | 15.53 |
| Mistral Large | 15.43 |
| GPT-4o | 13.24 |
| Gemini 2.5 Pro | 12.98 |
| Claude Sonnet 4.5 | 11.55 |
| Claude 3.7 + Reasoning | 10.43 |
| Claude 3.7 Sonnet | 8.51 |
| **Average** | **16.00** |

### S1.3.2 Damage Index

Finally, we calculate the **Damage Index** ($d_{m,f} = p_{m,f} \times \Delta_{m,f}$), which represents the expected accuracy loss (pp) per question attributable to failure $f$ for model $m$. Intuitively, $d_{m,f}$ combines "how often it happens" with "how much it hurts when it happens."

Table S4 summarizes the damage index across models. Three patterns stand out:

1. DISTRACTOR-LATCH causes the highest expected loss among evaluated failure modes for most models, signaling the need for stronger anchor discipline and distractor filtering late in the chain.

2. COVERAGE-GAP is the second critical factor; some models like *Llama 3.3 70B* (6.01 pp), *DeepSeek R1* (5.11 pp), and *GLM 4.6* (5.07 pp) are sensitive to this factor, while others like *Claude Sonnet 4.5* (2.00 pp) and *Grok 4 Fast* (1.55 pp) are more resilient, recovering with minimal hops or parametric knowledge.

Table S2: **Prevalence** $p_{m,f}$ of each failure (% of runs).

| Model | Coverage Gap (%) | Overconfident (%) | Distractor Latch (%) |
|---|---|---|---|
| Claude 3.7 Sonnet | 9.2 | 3.2 | 20.4 |
| Grok 4 Fast | 12.7 | 17.9 | 16.7 |
| Gemini 2.5 Pro | 13.9 | 15.5 | 19.1 |
| Mistral Large 2402 | 15.6 | 12.6 | 25.8 |
| GPT-5 | 29.2 | 22.9 | 17.2 |
| Llama 3.3 70B Instruct | 27.4 | 15.8 | 24.2 |
| Claude 3.7 Sonnet + Reasoning | 10.7 | 4.5 | 19.1 |
| GLM 4.6 | 16.4 | 29.6 | 16.4 |
| DeepSeek R1 | 18.4 | 33.9 | 21.6 |
| Claude Sonnet 4.5 | 13.6 | 11.5 | 15.0 |
| GPT-4o | 14.0 | 35.0 | 22.1 |

Table S3: **Impact** $\Delta_{m,f}$ of each failure (accuracy drop in percentage points; higher is worse).

| Model | Coverage Gap (pp) | Overconfident (pp) | Distractor Latch (pp) |
|---|---|---|---|
| Claude 3.7 Sonnet | 38.9 | 21.5 | 53.5 |
| Grok 4 Fast | 12.2 | 21.3 | 43.6 |
| Gemini 2.5 Pro | 30.8 | 9.0 | 60.8 |
| Mistral Large 2402 | 31.2 | 23.8 | 47.9 |
| GPT-5 | 15.8 | 15.6 | 58.6 |
| Llama 3.3 70B Instruct | 21.9 | 24.0 | 53.8 |
| Claude 3.7 Sonnet + Reasoning | 28.7 | 0 | 46.1 |
| GLM 4.6 | 31.0 | 23.1 | 51.8 |
| DeepSeek R1 | 27.7 | 8.2 | 55.0 |
| Claude Sonnet 4.5 | 14.7 | 11.1 | 54.4 |
| GPT-4o | 24.4 | 13.7 | 52.2 |

3. OVERCONFIDENCE (finalizing too early without enough coverage/sufficiency) negatively impacts some models (e.g., *GPT-5*: 3.58 pp; *GLM 4.6*: 3.39 pp), whereas models like *Claude 3.7 + Reasoning* show a near-zero damage index, indicating that their occasional early finalizations do not systematically reduce accuracy.

Overall, the observed damage indices suggest incorporating extended controller strategies: (i) reducing distractor latch via coverage-gated retrieval and anchor tracking; (ii) closing coverage gaps earlier for models with high coverage damage; and (iii) installing stricter early-stop guards for models prone to overconfidence.

### S1.4 Query Characteristics

To understand the qualitative differences in how models navigate the search space, we analyze the semantic properties of the queries they generate. We classify each retrieval query using four mutually exclusive quality flags: **Vague** (lacking concrete targets), **Over-Broad** (scope is too wide), **Off-Topic** (targets a subject not required by any oracle hop), and **Fusion** (attempts to solve multiple oracle hops simultaneously). Figure S3 combines our analysis of the accuracy impact (Panel a) and model prevalence (Panel b).

Our analysis of **Fusion** queries reveals a surprisingly mild impact on performance. As shown in Figure S3a, the accuracy gap between runs with and without fusion is relatively small (78.3% → 72.3%). This suggests that modern retrievers are increasingly capable of handling compound queries. Figure S3b confirms that high-performing reasoning models like DeepSeek R1 and Claude Sonnet 4.5 utilize fusion frequently (> 20% of steps), effectively using it as an efficiency strategy.

In contrast, **Over-Broad** queries impose a consistent penalty, dropping performance from 78.0% to 70.5%, likely due to noisy contexts diluting the signal. Panel b highlights that non-reasoning models generate over-

Table S4: **Damage index** $d_{m,f}$ (expected pp lost per question) for each model and failure. Higher is worse. Composition failure is omitted by design as it only manifests on incorrect answers.

| Model | Coverage Gap | Overconfident | Distractor Latch |
|---|---|---|---|
| Claude 3.7 Sonnet | 3.6 | 0.7 | 10.9 |
| Grok 4 Fast | 1.5 | 3.8 | 7.3 |
| Gemini 2.5 Pro | 4.3 | 1.4 | 11.6 |
| Mistral Large 2402 | 4.9 | 3.0 | 12.4 |
| GPT-5 | 4.6 | 3.6 | 10.1 |
| Llama 3.3 70B Instruct | 6.0 | 3.8 | 13.0 |
| Claude 3.7 Sonnet + Reasoning | 3.1 | 0 | 8.8 |
| GLM 4.6 | 5.1 | 3.9 | 8.5 |
| DeepSeek R1 | 5.1 | 2.2 | 11.9 |
| Claude Sonnet 4.5 | 2.0 | 1.3 | 8.2 |
| GPT-4o | 3.4 | 0.9 | 11.5 |

broad queries less frequently than models like Mistral Large or GPT-4o. **Vague** queries represent a clearer failure of intent formulation, leading to a significant accuracy drop ($77.5\% \rightarrow 66.0\%$). Finally, **Off-Topic** queries are the most detrimental, reducing accuracy to 59.2%. Although rare ($< 4\%$ for most models), GPT-5 and Gemini 2.5 Pro exhibit slightly higher rates of off-topic drift.

### S1.5 Case Studies and System Prompts

This section provides concrete examples of the failure modes discussed in the main text (Figures S4, S5, S6) and the exact system prompts used for our diagnostic suite (Figures S8, S9, S10).

### S1.6 Dataset Selection and Generalization Considerations

A central limitation of this study is that the main experiments are conducted on a single benchmark, ChemKGMultiHopQA. To evaluate whether additional datasets could support the same controlled comparison, we reviewed recent multi-hop QA benchmarks and assessed whether they satisfied the requirements needed for our three-regime evaluation: No Context, Gold Context, and Iterative RAG. The goal of this screening was not to exhaustively survey all multi-hop QA datasets, but to identify benchmarks that would allow a fair and interpretable test of whether synchronized retrieval and reasoning can outperform a static oracle-evidence condition.

We used four main criteria. First, the dataset should be recent and retrieval-dependent, so that strong LLMs are less likely to answer a large portion of the questions from parametric memory alone. This is important because high No-Context accuracy would weaken the interpretation of retrieval-augmented performance. Second, the dataset should use short, well-defined answers. Short-answer evaluation reduces ambiguity and allows more reliable correctness verification across models, especially when answers correspond to entities, chemical names, formulas, or other canonical identifiers. Third, the dataset should provide substantive gold contexts or supporting chunks. For the Gold Context condition to serve as an idealized static RAG baseline, the oracle evidence should contain realistic passages rather than minimal answer-bearing snippets. Fourth, the dataset should be scientific or domain-specific and corpus-based, since our goal is to study retrieval-dependent reasoning under specialized knowledge conditions rather than broad parametric recall over open-domain facts.

In addition to these criteria, our diagnostic analysis requires access to intermediate hops, sub-questions, or explicit reasoning paths. These annotations are necessary for measuring process-level behavior such as hop coverage, late-hop failures, anchor propagation, and evidence sufficiency. This requirement further narrows the set of compatible benchmarks. Table S5 summarizes the recent datasets we considered and the criteria they satisfy.

Table S5: Dataset-selection analysis for recent multi-hop QA benchmarks. "Recent/retrieval-dependent" indicates whether the benchmark is recent and likely to require retrieval rather than parametric recall. "Gold chunks" indicates whether substantive supporting context is available for constructing a Gold Context condition. "Domain corpus" indicates whether the benchmark is grounded in a scientific or domain-specific corpus. None of the reviewed alternatives simultaneously satisfied all requirements needed for our controlled comparison and mechanism-level diagnostics.

| Dataset | Recent | Short ans. | Gold chunks | Domain corpus |
|---|---|---|---|---|
| DocHopQA (Park et al., 2025b) | ✓ | – | ✓ | ✓ |
| BioCDQA (Feng et al., 2025) | ✓ | – | ✓ | ✓ |
| Koblex (Lee et al., 2025b) | ✓ | – | – | ✓ |
| BioHopR (Kim et al., 2025) | ✓ | – | – | ✓ |
| Cofca (Wu et al., 2024) | ✓ | ✓ | ✓ | – |
| ChemLit-QA (Wellawatte et al., 2024) | ✓ | – | – | ✓ |
| NovelHopQA (Gupta et al., 2025) | ✓ | ✓ | ✓ | – |
| Grade (Lee et al., 2025a) | ✓ | ✓ | – | – |
| MedHopQA (National Library of Medicine BioNLP Group, 2025) | ✓ | ✓ | – | ✓ |

As shown in Table S5, several datasets satisfy some of the desired properties, but none meet all of them simultaneously. Some recent scientific or biomedical datasets provide relevant corpora but do not use short, easily verifiable answers. Others provide short answers but lack substantive gold-context chunks or are not grounded in a specialized scientific corpus. In several cases, the available gold evidence is too short to approximate a realistic static RAG setting, reducing the task to answer extraction rather than multi-hop synthesis.

ChemKGMultiHopQA was therefore selected because it best satisfies the requirements of this study. It is recent, chemistry-focused, corpus-based, and contains one- to four-hop questions with short answers. It also provides oracle evidence and intermediate reasoning structure, enabling both the controlled comparison across No Context, Gold Context, and Iterative RAG and the mechanism-level diagnostics reported in Section 5. Nevertheless, the use of a single benchmark limits the generality of our conclusions. We therefore interpret our findings as evidence for scientific chemistry-domain multi-hop QA and leave broader validation across future compatible benchmarks as an important direction for future work.

### S1.7 Gold Context Length and Iterative RAG Improvements

Because Iterative RAG uses more output tokens than the static Gold Context condition, it is important to separate two possible explanations for its gains. One possibility is that Gold Context fails mainly because some oracle contexts are too long, creating context overload for the generator. Another possibility is that the advantage of Iterative RAG comes from the staged organization of evidence: the model retrieves, updates its partial answer state, and conditions each subsequent step on a narrower evidence set. To examine this issue, we analyze the relationship between Gold Context token length and accuracy.

Figure S13 groups questions into seven Gold Context token-count bins and reports the average Gold Context accuracy across all 11 evaluated models. The plot shows a mild downward trend: questions with shorter Gold Contexts are answered correctly more often, with the 0–150 token bin reaching approximately 80% accuracy. Accuracy generally decreases as Gold Context length increases. However, the large error bars indicate that this trend is not statistically decisive. In particular, among the bins with larger sample sizes, Gold Context length does not appear to substantially determine model accuracy. The sparsely populated longest-context bins should also be interpreted cautiously, especially the 1200–2000 token bin, which contains only a small number of questions.

Figure S14 further compares the Gold Context token-length distribution for two groups of questions: those answered correctly in the Gold Context condition, and those answered incorrectly in Gold Context but correctly in Iterative RAG. If Iterative RAG mainly helped because Gold Context inputs were too long, we would expect the improved questions to be concentrated in the longest-context region. Instead, the

two distributions overlap substantially across the full token-length range. Many Iterative RAG-improved questions fall in the same 150–800 token region as questions already solved by Gold Context. This indicates that Iterative RAG does not only recover failures from extremely long oracle contexts.

Together, these results suggest that Gold Context length contributes to difficulty but does not fully explain the gap between Gold Context and Iterative RAG. The advantage of Iterative RAG is therefore not merely a length effect. Rather, staged retrieval appears to help by controlling how evidence is introduced, maintaining a partial answer state, and aligning each retrieval step with the model's evolving reasoning trajectory.

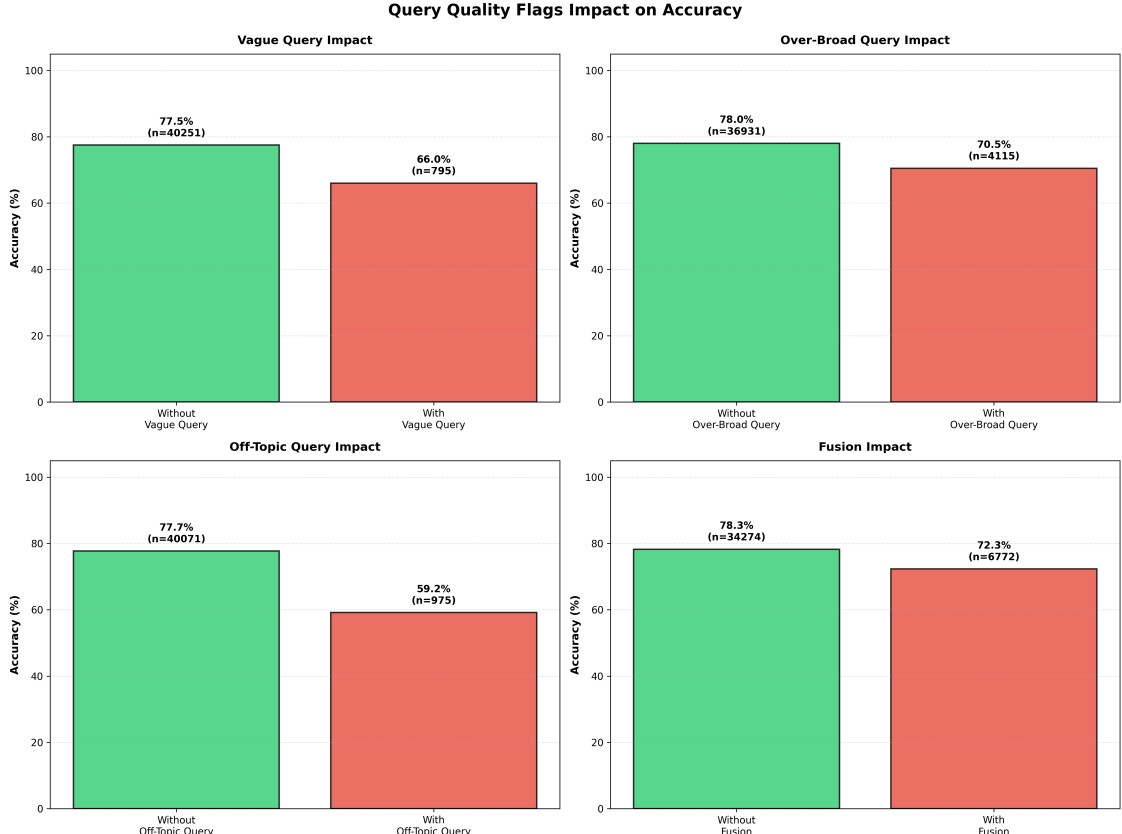

(a) **Impact on Accuracy:** Vague and Off-Topic queries cause the steepest drops (> 10 pp).

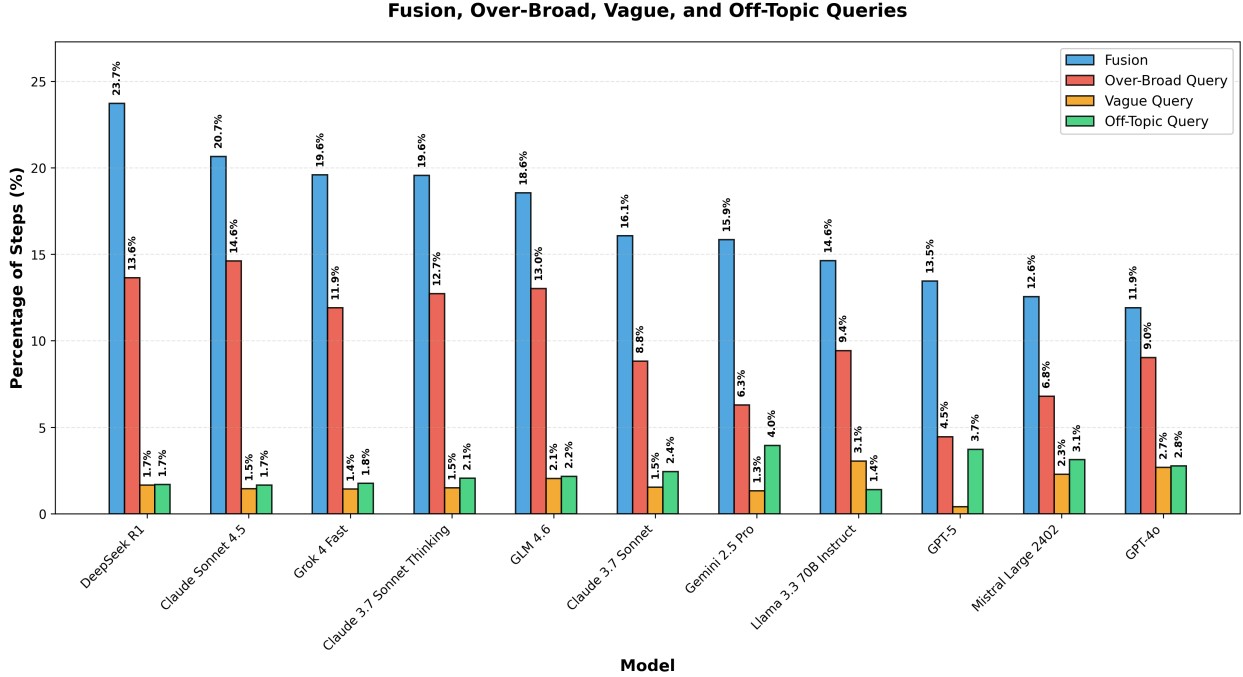

(b) **Prevalence by Model:** Reasoning models (left) favor Fusion; Standard models (right) prone to Over-Broad.

Figure S3: **Query Quality Analysis.** We diagnose query intent using four flags. Panel (a) shows that while Fusion is benign (small accuracy drop), Vague and Off-Topic queries are destructive. Panel (b) reveals that reasoning-optimized models leverage Fusion as an efficiency tool, whereas standard models struggle with Over-Broad queries.

---

**Case Study 1: Retrieval Coverage Gap (The "Broken Bridge")**

**Multi-hop Question:** *"What type of inorganic or organometallic compound, characterized by repeating coordination complexes... yields an inherently chiral **helicene-based molecular architecture**... eventually reacts with electrically conductive species known for forming robust **Schiff base complexes**?"*

**Oracle Reasoning Path (3 Hops):**

[**Hop 1**] **Start:** Identify inorganic compounds with repeating units.
→ **Entity:** Coordination Polymers

[**Hop 2**] **Bridge:** Identify the specific chiral architecture yielded by these polymers.
→ **Entity:** 3(2-pyridyl)-4-aza[6]helicene

[**Hop 3**] **End:** Identify species reacting with Schiff bases.
→ **Entity:** Metal Ions (Transition Metals)

---

**System Diagnosis:** While the model correctly identified the start (Coordination Polymers) and end (Metal Ions/Schiff Bases) of the chain, it suffered a complete **Coverage Gap at Hop 2**.

- **Step 1 Retrieval:** Successfully fetched definitions of "Coordination Polymers." (Covered)

- **Step 2 Retrieval:** Successfully fetched documents on "Schiff base ligands" and "Transition Metals." (Covered)

- **The Gap:** Zero documents were retrieved mentioning *"3(2-pyridyl)-4-aza[6]helicene"* or the specific helicene architecture. (Failed)

**Impact:** The model generated the correct final answer (*"A coordination polymer"*) by relying on internal parametric knowledge to leap over the missing bridge. However, the result is hallucinated verification: the system cannot prove *why* this specific polymer yields chirality because it missed the document describing the helicene unit.

---

Figure S4: **Example of Retrieval Coverage Gap.** The system successfully retrieves evidence for the start and end of the reasoning chain but fails to fetch the connecting document (Hop 2), resulting in a "broken bridge" in the evidence path despite a correct final answer.

---

**Case Study 2: Composition Failure (Precision Drift)**

**Question:** *"In what field of study would researchers be investigating a scenario where a uranium-based oxycation... induces in situ formation of peroxide species...?"*

**The Evidence Landscape:** The model successfully retrieved two key pieces of information:

- **Evidence A (Target):** "The uranyl cation has long been central to **uranium chemistry** research..." (Retrieved)

- **Evidence B (Distractor):** "...fruitful area of study for **actinide chemists** for decades." (Retrieved)

---

**System Diagnosis:** Although the specific answer (*"uranium chemistry"*) was present in the retrieved context, the model failed to synthesize it faithfully.

- **Retrieval Performance:** 100% Coverage. The system found the exact sentence answering the question.

- **Reasoning Trace:** The model vacillated between the field name and the researcher title.

- **Final Answer:** *"The field of study is **actinide chemistry**..."* (Composition Failure)

**Impact:** This represents a **Precision Drift**. The model allowed a related entity (*actinide chemists*) to override the explicit entity requested by the question (*uranium chemistry*). While semantically close, in specialized domains, substituting a hypernym (Actinide) for the specific field (Uranium) constitutes a failure to respect the retrieved evidence.

---

Figure S5: **Example of Composition Failure.** The model retrieves the correct evidence (Evidence A) but is distracted by adjacent concepts (Evidence B), leading to an imprecise final answer despite perfect retrieval.

---

**Case Study 3: Distractor Latch (The "Scaffold Trap")**

**Question:** *"Which class of ligands, **alongside** compounds known for forming Weitz type redox derivatives, is instrumental in modulating the performance of catalysts…?"*

**The Logic Trap:** The question asks for the *partner* of the entity described.

    **Context Entity:** Compounds forming Weitz derivatives → **Organophosphorus/Phosphines**.

    **Target Entity:** The class appearing *alongside* them → **N-heterocyclic carbenes (NHCs)**.

---

**System Diagnosis:** The retrieval system found the exact answer in Step 1 but became "latched" onto the distractor scaffold.

- **The Golden Snippet:** One retrieved text explicitly stated: *"…influenced by auxiliary ligands, such as **organophosphorus compounds** and **Nheterocyclic carbenes (NHCs)**."*

- **The Latch:** Despite having the answer, subsequent retrieval steps obsessed over "Phosphines" and "Schiff bases," flooding the context with information about the *context entity* rather than the *target*.

- **Final Answer:** *"Phosphine ligands are instrumental…"* (False Latch)

**Impact:** This illustrates a **Scaffold Trap**. The model failed to process the relational operator ("alongside"). Instead of solving for $X$ in "$X$ and $Y$," it simply recognized the features of $Y$ (Phosphines) and output $Y$ as the answer, driven by the high volume of retrieved text discussing Phosphine catalysis.

---

Figure S6: **Example of Distractor Latch.** The model successfully retrieves the source text containing the answer (NHCs) and the distractor (Phosphines). However, it fails to parse the specific relationship ("alongside"), getting trapped by the high frequency of the distractor terms in the evidence stream.

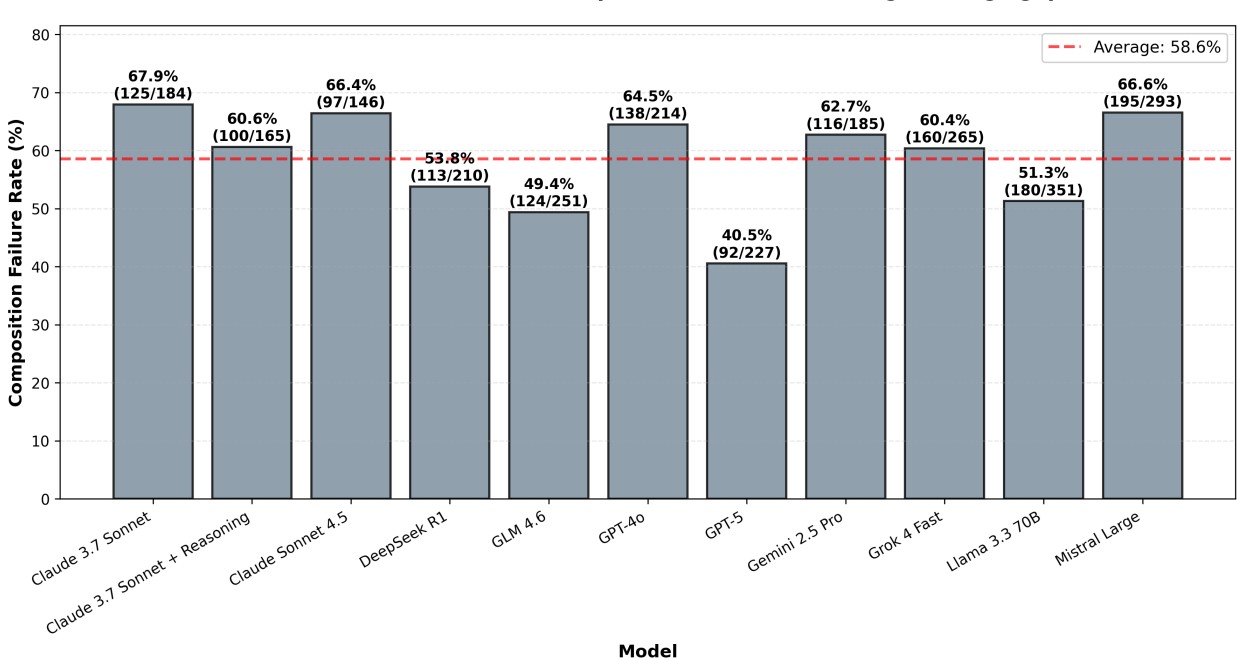

Figure S7: Strict Composition Failure Rates (Excluding Coverage Gaps). This plot shows the percentage of incorrect answers where the model retrieved **all** required reasoning hops (zero coverage gaps) yet still failed to produce the correct answer. The persistence of high failure rates (Average 58.6%) even when the full reasoning chain is present highlights the distinct difficulty of multi-hop synthesis independent of retrieval success.

---

**System Prompt: Faithfulness, Composition, and Confidence Auditor**

**Role Instructions:** You are an exacting auditor of an iterative retrieval–augmented QA system. Your job: judge the FINAL ANSWER for faithfulness to the provided evidence, detect composition failure, and diagnose confidence miscalibration. Use ONLY the supplied text. No outside knowledge. Be conservative when unsure. Return EXACT JSON in the schema below. No prose outside JSON.

**Scope of Judgment (Run-Level, Final Answer Focus):**

- **(1) Composition / Answer Synthesis Failure**
  - **True** if the correct entity/claim is present in the evidence but the final candidate ("candidate" key) either: (a) selects a different entity, (b) paraphrases without clearly naming the correct entity, or (c) muddles/merges entities so the core answer is wrong or unclear.
  - Note: `expected_answer` is the oracle answer for the full question.

- **(2) Unsupported Claim (Faithfulness)**
  - For each atomic sentence in the partial answers, decide if at least one evidence text supports it. Look for previous and current evidence texts in source steps.
  - **Support** = directly stated or a tight paraphrase; speculation is unsupported.

- **(3) Confidence Miscalibration**
  - **Purpose:** Detect when the system (i) answered confidently with weak/insufficient evidence, or (ii) kept retrieving despite already having enough evidence.
  - **Inputs:** `source_step`, `max_steps`, `num_hops`, `partial_answers`, `source_query`, `hop_subq`, `answer_subq`.
  - **Internal Estimates:**
    * `sufficiency_score_est` $\in [0,1]$: fraction of partial answer sentences supported by $\geq 1$ snippet.
    * `hop_coverage_est` $\in [0,1]$: fraction of oracle hops whose key entity/relation appears in partial answers OR evidence snippets.
  - **Decision Rules:**
    * *Overconfident Finalize (Under-thinking):* Trigger if (`finalize_step` < `num_hops`) AND (`hop_coverage` < 0.8 OR `sufficiency` < 0.60).
    * *Underconfident Continue (Over-thinking):* Trigger if some prior step $t <$ `finalize_step` likely had "enough" evidence to support the final answer.

Figure S8: System prompt used for evaluating Faithfulness, Composition Failure, and Confidence Calibration.

---

**System Prompt: Query Quality and Distractor Diagnosis Auditor**

**Role Instructions:** You are an exacting auditor of an iterative retrieval–planning RAG system. For EACH step, judge the step's intended hop and the quality of its query. Also detect partial-answer contradictions across steps, and a run-level "distractor latch". Use ONLY the provided text. Return EXACT JSON.

**Judgments to Make:**

- **(1) Next-Logical-Hop (Hop Intent)**
  - `predicted_hop`: Which oracle hop the query primarily aims to solve (1-based). Match surface forms against hop entities.
  - `is_next_logical_hop`: True iff `predicted_hop` == (`resolved_hops` + 1).
  - `fusion`: True if the query tries to solve multiple oracle hops at once.
- **(2) Query Quality Flags**
  - `vague`: Lacks concrete targets (e.g., "learn more about HAT").
  - `over_broad`: Scope is too wide or mixes unrelated facets.
  - `compound`: Bundles multiple sub-questions with AND/OR.
  - `off_topic`: Targets a subject not required by any oracle hop.
  - `anchored`: Includes at least one salient anchor from the previous partial answer (e.g., "Fe(IV)=O", "H2"). Ignore generic words.
  - `hallucinated_term`: Contains specific constraints/names NOT present in history or evidence (ignore for step 1).
- **(3) Partial Contradiction (Step $t \geq 2$)**
  - `partial_contradiction_with_prev`: True if `partial_answer_t` conflicts with `partial_answer_(t-1)` (mutually exclusive claims).
- **(4) Distractor Latch / Scaffold Trap (Run Level)**
  - **Definition:** True if retrieved evidence is locked onto a chemically similar but irrelevant scaffold/family compared to the gold target (e.g., "phenyl" vs needed "phenoxyl").
  - **Method:** Use simple family/entity-pattern matching over snippets vs. oracle hop entities. Be conservative.

**Operational Rules:** Use only provided text. Multiple flags can be true. Anchors/Hallucinations are false at step 1. Keep judgments conservative.

Figure S9: System prompt used for diagnosing Query Quality and Distractor Latch failures.

---

**System Prompt: Retrieval Coverage and Anchor Tracking Auditor**

**Role Instructions:** You are an exacting QA auditor for an iterative retrieval–planning–composition RAG system. Given one question, its oracle hop path, the system's per-step queries/partial answers, and retrieved snippets, return concise JSON labels for THREE diagnostics only.

**Diagnostics Definitions:**

- **(1) Retrieval Coverage Gap (missed-hop)**
  - **Definition:** For any oracle hop $k$, across ALL steps, NONE of the retrieved snippets are about that hop's key entity/relationship.
  - **Meaning:** The system never fetches the document(s) needed for one of the hops.
  - **Output:** List of `missed_hops` (by index) + overall boolean.
- **(2) Anchor Carry-Drop**
  - **Definition:** If at step $t > 1$ the previous partial answer names a key entity/anchor, the query at step $t$ SHOULD carry at least one of those anchors.
  - **Logic:** If it carries none, the step is a carry-drop. Only judge when a previous partial exists and has salient anchors.
  - **Output:** Per-step true/false and overall boolean.
- **(3) Late-Hit per Hop**
  - **Definition:** For oracle hop $k$, find the FIRST step where any snippet contains that hop's key entity.
  - **Logic:** If `first_hit_step_for_hop_k > hop_index`, mark `late_hit=true`.

**Rules & Heuristics:**

- Work only with supplied text. No world knowledge beyond common-sense aliasing.
- "Snippet mentions hop entity" = snippet text or query text includes clear surface form of hop's key entity.
- Anchor detection: anchors are explicit proper names, formulae, or distinctive class labels. Ignore generic words (e.g., "catalyst").
- Be conservative: prefer false over true when ambiguous.

Figure S10: System prompt used for evaluating Retrieval Coverage gaps and Anchor Carry-Drop rates.

---

**System Prompt: Chemical Entity Equivalence Verifier**

**Task:** Decide whether Expected and Candidate name the SAME chemical entity.

**What counts as the SAME:**

- Aliases, common vs IUPAC names, and formulas refer to the same thing (e.g., lithium chloride = LiCl; acetic acid = ethanoic acid).
- Minor packaging/context words do not change identity: material, compound, sample, reagent, powder, nanopowder, precursor, solution.
- The Candidate may be a long sentence or paragraph with explanations; as long as it explicitly names the same entity as the answer, count it as the same.

**What is NOT the same:**

- Different polymorph/crystal structure/phase (wurtzite ZnO vs rocksalt ZnO).
- Different charge state or ion vs neutral; cation vs anion (Li vs Li+; chloride ion vs HCl).
- Different oxidation state or stoichiometry (FeCl2 vs FeCl3).
- Different hydration/solvation (CuSO4 vs CuSO4*5H2O).
- Different stereochemistry or isotopic labeling (L- vs D-; 13C-labeled vs unlabeled).
- Salt vs parent acid/base (acetate vs acetic acid).
- Class/family vs specific member (alkali metal chloride vs lithium chloride) unless the specific Expected entity is explicitly named.
- Candidate only mentions Expected to negate or contrast it (uses words like 'not', 'instead of', 'different from', 'vs') while naming a different main entity.

**Decision rule:**

- If Candidate explicitly names the same entity as Expected (even inside a longer explanation), answer: **true**.
- Otherwise, answer: **false**.

**Output:** Answer with exactly: **true** or **false**.

**Examples:**

- **Expected:** wurtzite ZnO
  **Candidate:** The ZnO polymorph used as the precursor in the synthesis of rsZnO according to high-pressure nanopowder synthesis methods is wurtzite ZnO (wZnO).
  **Answer:** true
- **Expected:** wurtzite ZnO
  **Candidate:** The product was rocksalt ZnO (rs-ZnO), not wurtzite ZnO.
  **Answer:** false

Figure S11: System prompt used for verifying the correctness of the final answer by comparing the generated candidate against the expected oracle entity.

---

**System Prompt: Gold Context + Chain-of-Thought Baseline**

**Role:** You are an expert for multi-hop Question Answering over unstructured text in chemistry.

**Output format:** Return ONLY a valid JSON object. No prose outside the JSON. No markdown formatting outside the JSON block. No comments.

**Required schema:**

$$\{\text{"CoT"} : \text{"..."}, \text{"answer"} : \text{"..."}\}$$

**Inputs provided:**

- **question:** The full multi-hop user question.
- **paragraphs:** The text passages containing the potential evidence.

**Policy:** You must document your step-by-step reasoning in the `CoT` field before providing the final `answer`. Follow these strict steps within `CoT`:

- **Decompose:** Explicitly break down the multi-hop `question` into an ordered list of atomic, single-hop sub-questions and explain why you are breaking it down in such a way.
- **Extract & Answer:** Go through each sub-question one by one. Search the provided `paragraphs` for the evidence needed.
- **Strict Grounding:** Never use outside knowledge.
- **Synthesize:** Based on the extracted evidence, determine the final answer.
- **Final Output:** Place your final, concise conclusion in the `answer` field.

**Required CoT sections:** The `CoT` field must be highly detailed and explicitly include the following sections:

- **[Hypothesis Formulation]:** State detailed initial thoughts on how to approach the sub-questions.
- **[Verification]:** Double-check if the extracted quotes fully satisfy the sub-questions without making assumptions.

---

Figure S12: Main system prompt used for the Gold Context + Chain-of-Thought baseline. The prompt gives the model the same oracle evidence as the Gold Context setting, but explicitly asks it to decompose the question, extract evidence step by step, verify grounding, and return a JSON object with reasoning and final answer fields.

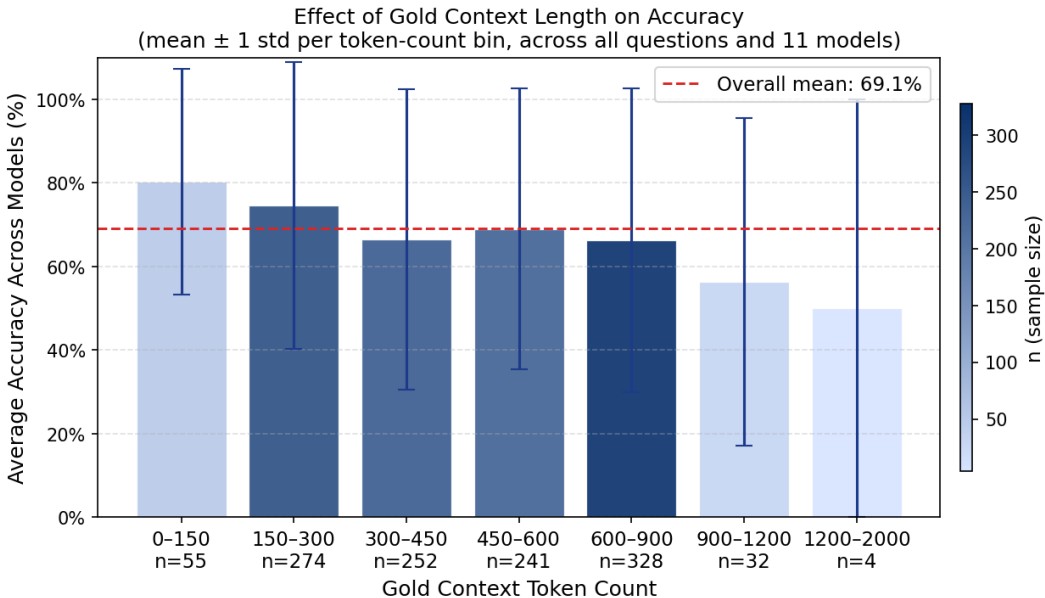

Figure S13: Effect of Gold Context length on Gold Context accuracy. Questions are grouped into token-count bins, and each bar reports the average accuracy across all 11 models. Error bars show one standard deviation, and $n$ denotes the number of questions in each bin. Accuracy decreases mildly as Gold Context length increases, but the trend is not statistically decisive across the well-populated bins.

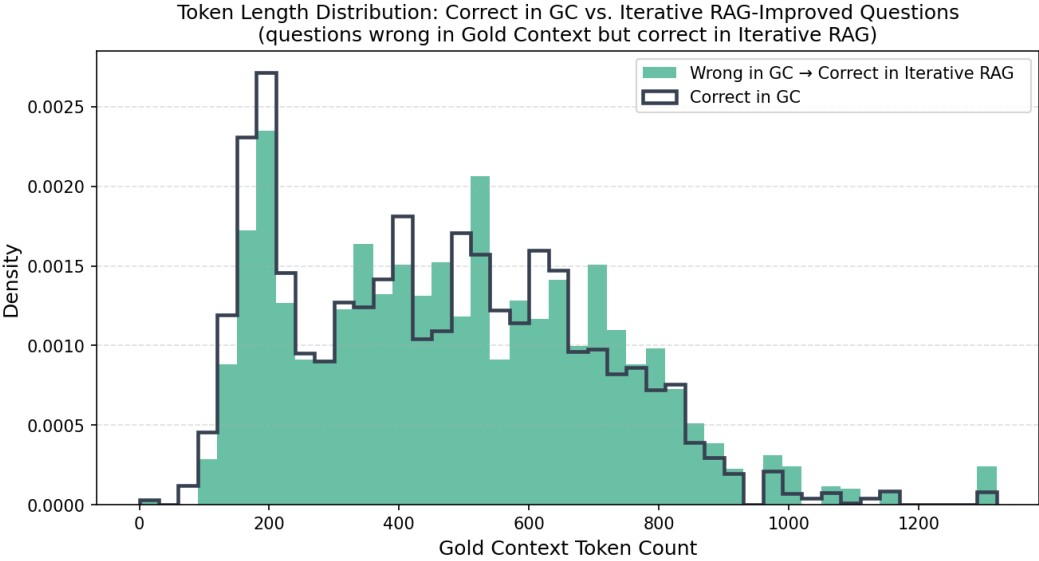

Figure S14: Token-length distribution of Gold Contexts for questions answered correctly in Gold Context versus questions answered incorrectly in Gold Context but correctly in Iterative RAG. The substantial overlap between the two distributions suggests that Iterative RAG improvements are not limited to unusually long Gold Contexts. Instead, improvements occur across a broad range of context lengths.

