# OpenReview forum: "When Iterative RAG Beats Ideal Evidence: A Diagnostic Study in Scientific Multi-hop Question Answering"
_TMLR — Accepted by TMLR_

### Review · Reviewer_NeJD · 2026-02-12

**Summary Of Contributions:**

This paper presents a study evaluating the performance of RAG in scientific multi-hop question answering. The authors benchmark eleven LLMs (including GPT-4o, Claude 3.7, and DeepSeek R1) across three regimes: No Context, Gold Context (oracle evidence), and Iterative RAG. Using the ChemKGMultiHopQA dataset, the study challenges the assumption that providing "ideal evidence" (Gold Context) is the upper bound for performance.

The key finding is that Iterative RAG consistently outperforms Gold Context, with gains of up to 25.6 percentage points, particularly for non-reasoning models.

The authors attribute this to the cognitive benefits of staged retrieval and reasoning, despite identifying failure modes like "distractor latching" and "composition failure".

**Audience:**

Yes

**Audience Explanation:**

Please refer to the comments above, divided into strengths and weaknesses.

**Claims And Evidence:**

No

**Claims Explanation:**

**Strengths:**

1.The paper provides compelling evidence that the process of retrieval is often more valuable than the result (static evidence). The finding that Iterative RAG beats the theoretical "Gold Context" upper bound is a significant contribution that challenges standard RAG evaluation paradigms.

2.The authors go beyond simple accuracy metrics. The introduction of specific metrics like "Retrieval Coverage Gap," "Anchor Carry-Drop," "Procedural Compliance Rate (PCR)," and "Distractor Latch" provides a granular understanding of why models fail.

**Weakness:**

1. *Single-Domain Limitation:* While the focus on Chemistry is a strength for reducing data leakage, it limits the claim of generalizability. The complexity of chemical nomenclature and the specific nature of "distractor latches" in chemistry (e.g., similar molecular structures) might not translate perfectly to other domains like math, medical, legal or financial QA.

2. *Fairness of Comparison (Compute & Latency):* The comparison regarding computational cost is highly asymmetric. As shown in Table 1, the token consumption for Iterative RAG is vastly higher than the Gold Context baseline. For instance, Claude 3.7 Sonnet (Reasoning) output increases from an average of ~715 tokens in the Gold setting to ~4,309 tokens in the Iterative setting. Similarly, GPT-4ojumps from just ~9 tokens (likely direct answers) in the Gold setting to ~448 tokens in the Iterative setting. The authors explicitly note that Iterative RAG uses about three times the output tokens of the No Context baseline and requires up to 5 sequential retrieval steps. Comparing a multi-turn, high-compute process against a single-shot generation creates an unequal playing field, making it unclear if the gains stem from the iterative mechanism itself or simply the massive increase in compute budget allowed for reasoning.

3. *Ambiguity of "Gold Context" Optimality:* The paper argues Gold Context is an "idealized static upper bound". However, the high rate of "Composition Failure" (87.3% of incorrect answers) suggests the "Gold" baseline might not be optimized for how models actually consume context. If the Gold Context provided is too long or contains complex distractors within the gold paragraphs, the model's failure might be due to context length limits or attention span rather than a lack of reasoning capability.

**Requested Changes:**

1. **Compute-Normalized Baseline:**
To make the comparison fairer, add a "Chain-of-Thought (CoT) on Gold Context" baseline. Give the models the Gold Context but allow them to use a Chain-of-Thought reasoning process (with similar token usage to the Iterative RAG) before answering. This would isolate whether the gain comes from iterative retrieval or simply more reasoning time

2. **Deeper Analysis of Gold Context Failures:**
The high rate of "Composition Failure" (87.3% of incorrect answers) is alarming. A qualitative analysis of why models fail with Gold Context is needed. Are the gold passages too long? Do they contain conflicting information? This would clarify if the advantage of Iterative RAG is simply that it feeds the model smaller, more digestible chunks of text

3. **Ablation of the "Controller"**
A qualitative analysis of why models fail with Gold Context is needed given the high composition failure rate. Are the gold passages too long? Do they contain conflicting information? This would clarify if the advantage of Iterative RAG is simply that it feeds the model smaller, more digestible chunks of text to avoid context overload

---

> ### Author Response · Authors · 2026-03-04
> **Author Response to Reviewer NeJD**
>
> Thank you for your constructive review and for highlighting the significance of our findings. We have addressed your specific concerns below:
>
> **Requested Change. 1 Compute-Normalized Baseline (CoT on Gold Context)**
>
> We agree that a compute-normalized baseline adds valuable context to our evaluation. To address this, we conducted an ablation study implementing a "Gold+CoT" baseline for both a non-reasoning and a reasoning model (GPT-4o and Claude 3.7 Sonnet as requested). We implemented a compute-normalized "Gold+CoT" ablation, optimizing CoT prompts with few-shot examples to encourage step-by-step logic.
>
> | Model Name | Gold Accuracy | Gold Output Tokens | Gold+CoT Accuracy | Gold+CoT Output Tokens | Iterative RAG Accuracy | Iterative RAG Output Tokens |
> |---|---|---|---|---|---|---|
> | GPT-4o | 56.32% | 10 | 78.08% | 262.10 | **81.96%** | 448.63 |
> | Claude 3.7 Sonnet (Standard) | 68.13% | 30 | 81.45% | 909.96 | **84.49%** | 714.73 |
> | Claude 3.7 Sonnet (Thinking) | 73.27% | 715 | 83.22% | 2133.52 | **86.09%** | 4309.00 |
>
> Notably, Claude 3.7 Sonnet (Standard) used significantly more tokens in Gold+CoT (909.96) than in Iterative RAG (714.73), yet achieved lower accuracy (81.45% vs 84.49%). To maximize token usage in the Gold+CoT setup, we explicitly set the reasoning effort to "high" for Claude 3.7 Sonnet (Thinking), whereas all models in the Iterative RAG setup used a "medium" reasoning effort. Despite increasing compute as much as we could in CoT setup, Iterative RAG still achieved higher accuracies for both GPT-4o (81.96%) and Sonnet 3.7 Thinking (86.09%).
>
> Additionally, the higher token usage in Iterative RAG (where GPT-4o averages 3.93 iterations) is a natural architectural byproduct of generating search queries, passing partial answers, and queries and managing state. As visualized in Figure 8, adhering to a structured, multi-step process fundamentally drives accuracy, not sheer token consumption. Consequently, GPT-4o achieves better results in an Iterative RAG setup while using only 8% of the token budget required by larger models like GPT-5.
>
> **Requested Change 2 and 3. Gold Context Optimality & Failure Analysis**
>
> To clarify, the "composition failure" metric analyzes errors specifically within the Iterative RAG setup, and not GC. In the Gold Context baseline, the two dominant error modes are *Distractor Latching* (getting distracted by other provided text) and *Parametric Reliance* (hallucinating based on internal weights rather than provided passages).
>
> The Gold Context acts as an idealized, noise-free upper bound (averaging ~188, std: 56.26, tokens per passage; ~750 tokens for a 4-hop query). By contrast, our Iterative RAG system processes significantly more text---up to 10 passages per iteration (220 tokens each), plus previous best passages, queries, and partial answers---totaling over 10 times the volume of the Gold Context setup.
>
> In standard multi-hop queries, providing all oracle contexts upfront can cause the ideal paragraph for one hop to act as a distractor for another. Iterative RAG mitigates this by synchronizing retrieval and reasoning step-by-step to maintain model focus, proving performance gains are not simply due to longer output generation.
>
> As requested, we added a plot illustrating the effect of Gold Context length on average accuracy across all 11 models (overall mean: 69.1%). While we observe a slight downward trend as context length increases, starting at ~80% accuracy for the shortest contexts (0–150 tokens, n=55, where n is the number of questions), accuracy remains relatively stable across the most well-populated bins (150–900 tokens, n ≥ 241), fluctuating only between roughly 66% and 74%. The large, overlapping standard deviations across these core bins further indicate this minor variation is not statistically significant. Accuracy drops more noticeably for contexts exceeding 900 tokens, but these bins have small sample sizes (n=32 and n=4). Therefore, within standard operating lengths, Gold Context length alone does not substantially dictate model accuracy.
>
> **Single-Domain Limitation**
>
> As reflected in our title, this work focuses specifically on the scientific domain. We actively sought comparable multi-hop datasets in other domains that provide the intermediate hops and supporting evidence necessary for our in-depth evaluation; unfortunately, such comprehensive datasets are currently lacking. For example, we evaluated Koblex in the legal domain but found it unsuitable for two key reasons. First, the required answers were too long, making automated, style-agnostic verification unreliable. Second, its "gold contexts" were typically only one or two sentences long, which failed to provide the complex multi-hop conditions our benchmark explicitly tests and was incompatible with standard retrieval chunk sizes. We remain committed to searching for compatible, open-source multi-hop QA datasets and plan to expand this benchmark beyond the scientific domain in future work.

---

> > ### Author Response · Authors · 2026-03-17
> > **Request for Timely Feedback on Updated Revisions**
> >
> > Dear Reviewer,
> >
> > Thank you again for your productive feedback on our work. Since your last comments, we have updated the manuscript with the requested analyses, including the new **Gold+CoT ablation**, the clarification of **error modes** (Gold Context vs. Iterative RAG), and a **deeper context‑length analysis** accompanied by a new figure. These additions directly address the three points you raised and we hope they help clarify the underlying mechanisms behind the observed results.
> >
> > We would be grateful for any further insights or follow‑up questions you may have.
> > A response in your earliest convenience would allow us to provide additional clarification or analyses if needed before March 20 deadline.
> >
> > Thank you again for your time and constructive engagement.

---

> > > ### Comment · Reviewer_NeJD · 2026-04-01
> > >
> > > My questions are addressed.
> > > Thanks.

---

### Review · Reviewer_zUWy · 2026-03-02

**Summary Of Contributions:**

This paper demonstrates that a straightforward, training-free iterative RAG setup outperforms even idealized static Gold-Context RAG (where all oracle evidence is provided upfront) on retrieval-dependent multi-hop chemistry questions from the ChemKGMultiHopQA dataset. It reports consistent accuracy gains across 11 LLMs, attributing this to staged retrieval's ability to reduce late-hop failures, mitigate context overload, enable dynamic self-correction of hypothesis drift, and avoid distractor issues that static bulk evidence cannot address. The work's main value lies in its mechanism-level diagnostic framework, which audits failures via novel process signals like anchor carry-drop, procedural compliance rate, distractor latch, confidence miscalibration in stopping, query-quality flags, retrieval coverage gaps, and composition failures. It further profiles utilization dynamics, solvability partitions, cost-accuracy archetypes, and efficiency trade-offs, challenging the assumption that "more ideal evidence" suffices for complex scientific multi-hop QA and offering practical guidance for robust, controllable iterative RAG in specialized domains.

**Audience:**

Yes

**Audience Explanation:**

The work provides a controlled empirical/diagnostic study on Retrieval-Augmented Generation, specifically examining iterative vs. static RAG in scientific multi-hop QA. Key findings that iterative RAG outperforms idealized Gold-Context evidence, with mechanism-level breakdowns of why via diagnostics like anchor carry-drop, composition failures, and stopping miscalibration offer actionable insights for community building or evaluating RAG systems, especially in specialized domains.

**Broader Impact Concerns:**

Potential indirect risks such as improved RAG reliability enabling more convincing but still hallucinated scientific claims, or over-reliance on iterative systems in real-world research tools, are acknowledged in passing through discussions of persistent failure modes and the need for controllability.

**Claims And Evidence:**

Yes

**Claims Explanation:**

The key claims, numerical accuracies (e.g., aggregate averages of 37.16% for No Context, 69.14% for Gold Context, and 80.89% for Iterative RAG), gains (e.g., +25.64 pp for GPT-4o from Gold to Iterative), model specifics, and diagnostic details are directly supported and accurate.

**Requested Changes:**

- Figure 13: Overlapping content
- Wrong citation format:
  - "FinQA Chen et al. (2021) and TAT-QAYu et al. (2021)" → "FinQA (Chen et al., 2021) and TAT-QA (Yu et al., 2021)"
  - ensure consistent \citep and \citet style, proper spacing, no run-ons, and correct author-year separation.
  - page 9 "Llama 3 (et al., 2024)"
- Fix typos and phrasing issues
  - Figure 2 caption "Models’ accuracy Distribution of models’ accuracy in three setups" → "Models’ Accuracy: Distribution of Models’ Accuracy in Three Setups"
  - page 10 "(iii) Synchronized retrieval and reasoning Wins" → "(iii) Synchronized Retrieval and Reasoning Wins"
  - p22 "think longer" → ``think longer''
- Generalizability to other scientific or open-domain multi-hop settings remains untested.
- The work only adopts one specific iterative controller design (fixed budget, top-k retrieval, etc.): Results may be sensitive to variations in controller hyperparameters, query generation prompts, or more advanced agentic alternatives.

---

> ### Author Response · Authors · 2026-03-16
> **Author Response to Reviewer zUWy**
>
> Thank you for your positive review and for recognizing the value of our approach to dissecting iterative RAG failures. We are glad that our challenge to the ideal evidence assumption resonated with you. We have **addressed** all your requested changes to strengthen the manuscript, including making all requested formatting and typographical corrections throughout the text.
>
> Regarding the generalizability of our findings to other domains, we conducted an extensive search for suitable multi-hop QA datasets and applied a set of strict selection criteria, which we outline below along with the rationale for each. We also provide a list of recent (2025–2026) multi-hop QA datasets that we considered during this process.
>
> **1. Recency of the dataset.** Model performance on older benchmarks tends to be inflated due to data leakage into pretraining corpora. For instance, as reported by Khodadad et al. (2025), GPT-4o achieves 55% accuracy on HotpotQA in the No Context setting—substantially higher than on more recent benchmarks. Newer datasets are more likely to draw from corpora published after the knowledge cutoff of current LLMs, making them better suited for evaluating genuine retrieval dependence.
>
> **2. Domain specificity.** Scientific domains produce more challenging multi-hop questions that are less likely to appear in LLM training data. Many existing multi-hop datasets rely on Wikipedia as a corpus and pose questions that are too general to stress-test retrieval-dependent reasoning. For example, consider this question from the HotpotQA chemistry subset: *"Who is the father of Aage Bohr and son of Christian Bohr?"* Such questions can often be resolved through parametric memory alone. A scientific domain ensures that retrieval is genuinely necessary rather than merely supplementary.
>
> **3. Short-answer format.** Datasets with concise, well-defined answers enable more reliable and reproducible correctness verification. In ChemKGMultiHopQA, answers typically correspond to a specific chemical entity (e.g., a compound name, formula, or class), which makes it straightforward to determine whether the model's response is correct, independent of surface-level variation in phrasing.
>
> **4. Realistic gold context length.** In many existing datasets, the gold context for multi-hop questions—even those requiring 3 or 4 hops—consists of only 2–3 sentences that contain little beyond the answer itself. Under such conditions, the model does not need to reason through the provided paragraphs; it merely needs to extract the answer from a minimal, near-trivial context. In our study, Gold Context is intended to represent the upper bound of a static RAG system: the model receives one full paragraph per hop (since each hop originates from a distinct document), and each paragraph has an average length of 188 tokens. This ensures that the model must actively process and synthesize information from substantive passages rather than perform shallow extraction.
>
> Below is a list of recent multi-hop QA datasets (2025–2026) that we evaluated against these criteria.
>
> | Dataset | Recent  | Short Answer | Gold Context Chunk Available | Scientific Domain & Corpus Based |
> |---|---|---|---|---|
> | DocHopQA | ✅ | ❌ | ✅ | ✅ |
> | BioCDQA | ✅ | ❌ | ✅ | ✅ |
> | Koblex | ✅ | ❌ | ❌ | ✅ |
> | BioHopR | ✅ | ❌ | ❌ | ✅ |
> | Cofca | ✅ | ✅ | ✅ | ❌ |
> | ChemLit-QA | ✅ | ❌ | ❌ | ✅ |
> | NovelHopQA | ✅ | ✅ | ✅ | ❌ |
> | Grade | ✅ | ✅ | ❌ | ❌ |
> | MedHopQA | ✅ | ✅ | ❌ | ✅ |
>
>
> Regarding the sensitivity of our results to the controller design, your observation is entirely spot on. Our study utilized a specific, fixed-budget controller with top-k retrieval to establish a tightly controlled and uniform baseline across all eleven evaluated models. This iterative controller was built upon recent advancements in the literature, and its prompt was iteratively engineered to ensure a stable mechanism for this specific experiment. We absolutely acknowledge that there is room for further refinement of this controller. Future work could explore the impact of integrating a reranker, experimenting with different embedding models, or tuning other retrieval hyperparameters to see how they influence the models' iterative behavior. We have added a discussion of these variables to the manuscript to better contextualize our findings.

---

### Review · Reviewer_ZSY4 · 2026-03-06

**Summary Of Contributions:**

This paper presents a diagnostic study of iterative retrieval augmented generation for scientific multi-hop question answering. The authors evaluate eleven large language models under three inference regimes: No Context (parametric knowledge only), Gold Context (oracle evidence provided at once), and Iterative RAG, where retrieval and reasoning are interleaved through a training-free controller.

Experiments are conducted on the ChemKGMultiHopQA benchmark, a chemistry-domain multi-hop QA dataset. The paper introduces several diagnostic metrics to analyze model behavior, including retrieval coverage gaps, anchor carry-drop across reasoning steps, stopping calibration, and composition failures. The main claim is that staged retrieval, aligned with reasoning trajectories, can outperform a static setup in which all oracle evidence is provided simultaneously.

Strengths include the systematic comparison across three evaluation regimes, evaluation across a relatively large set of modern LLMs, and detailed diagnostic analyses of retrieval-reasoning interactions. Potential weaknesses include reliance on a single domain-specific dataset, limited comparison with existing iterative RAG approaches, and strong novelty claims regarding iterative retrieval outperforming oracle evidence.

**Audience:**

Yes

**Audience Explanation:**

The interaction between retrieval and reasoning in large language models is an active research area. The paper explores an interesting question: whether staged retrieval aligned with reasoning trajectories can outperform static setups that provide all evidence at once.

The diagnostic analysis of failure modes such as retrieval coverage gaps, composition failures, and stopping miscalibration may also be useful to researchers studying retrieval-augmented generation systems. However, the broader significance of the findings depends on whether the reported phenomenon generalizes beyond the specific dataset and controller design used in this study.

**Claims And Evidence:**

Yes

**Claims Explanation:**

The empirical study is extensive and well-organized. The authors evaluate 11 LLMs across three regimes and report consistent improvements when retrieval and reasoning are interleaved via an iterative controller, compared to both parametric inference and the Gold Context setup. Additional analyses, such as recoveries versus regressions, parametric suppression rates, and failure-mode breakdowns, provide useful insights into how iterative retrieval interacts with reasoning.

However, the evidence remains limited in scope. The experiments rely on a single dataset and a specific controller design, which makes it difficult to determine whether the observed improvements reflect a general property of iterative retrieval or depend on particular implementation choices. Some diagnostic metrics also rely on heuristic thresholds that may affect interpretation. Overall, the results support the empirical observation within this benchmark, but broader conclusions require further evidence.

**Requested Changes:**

### Generalization beyond a single dataset

The main experiments are conducted on ChemKGMultiHopQA. While this dataset is suitable for studying retrieval-dependent reasoning, it remains unclear whether the key finding that iterative retrieval can outperform Gold Context generalizes to other domains or benchmarks. Additional experiments or a clearer discussion of this limitation would strengthen the paper.

### Relationship to existing iterative RAG approaches

The paper evaluates a specific training-free controller but does not clearly establish why this design should be considered representative of iterative RAG systems. The relationship between the proposed setup and existing iterative retrieval-reasoning frameworks remains unclear. Comparisons with representative iterative RAG methods or stronger justification of the controller design would improve the validity of the conclusions.

### Scope of the novelty claim

The paper claims that it demonstrates for the first time that iterative RAG can outperform ideal oracle evidence in scientific multi-hop QA. While the results support this observation within the current setup, the novelty claim should be scoped more carefully relative to prior work on iterative retrieval and retrieval-reasoning loops.

### Justification of diagnostic metrics

Several diagnostic metrics rely on threshold-based definitions such as coverage or sufficiency levels. Providing justification or sensitivity analysis for these thresholds would improve robustness.

### Discussion of cost-accuracy trade-offs

Iterative retrieval substantially increases token usage and inference cost compared to static RAG. A clearer discussion of the cost-accuracy trade-off would improve the practical relevance of the work.

---

> ### Author Response · Authors · 2026-03-19
> **Author Response to Reviewer ZSY4**
>
> Thank you for your thorough and constructive review. We appreciate the careful attention to both the strengths and limitations of our work, and we have addressed each of your specific concerns below.
>
> **Generalization beyond a single dataset**
>
> We agree that evaluating iterative retrieval beyond ChemKGMultiHopQA would meaningfully strengthen the generalizability of our findings. Throughout this project, we conducted multiple rounds of searching (at project start, mid-study, and during final revisions) for additional recent, retrieval-dependent, multi-hop QA datasets. Unfortunately, we were unable to identify any other benchmark that satisfies the criteria required for a fair and interpretable comparison across No-Context, Gold-Context, and Iterative RAG conditions.
>
> To avoid overstating generalization, we have expanded the Discussion and Limitations sections to clearly articulate why ChemKGMultiHopQA was selected and to acknowledge that broader validation remains an open direction. We will also release the full code and evaluation pipeline to facilitate application of our method to future datasets that meet these criteria.
>
> Below we summarize the strict selection requirements and the rationale behind each:
>
> **1. Recency of the dataset.** Benchmarks must be recent enough that large language models still require retrieval to answer the questions. Older datasets often suffer from data leakage, inflating No-Context accuracy and distorting the comparison. For instance, as reported by Khodadad et al. (2025), GPT-4o reaches ~55% accuracy on HotpotQA in the No-Context setting. Recent datasets reduce this risk and produce a more meaningful retrieval–reasoning evaluation.
>
> **2. Scientific domain.** Scientific domains produce more challenging multi-hop questions that are less likely to appear in LLM training data. Many existing multi-hop datasets rely on Wikipedia as a corpus and pose questions that are too general to stress-test retrieval-dependent reasoning. For example, consider this question from the HotpotQA chemistry subset: *"Who is the father of Aage Bohr and son of Christian Bohr?"* Such questions can often be resolved through parametric memory alone. A scientific domain ensures that retrieval is genuinely necessary rather than merely supplementary.
>
> **3. Short-answer format.** Datasets with concise, well-defined answers enable more reliable and reproducible correctness verification. In ChemKGMultiHopQA, answers typically correspond to a specific chemical entity (e.g., a compound name, formula, or class), which makes it straightforward to determine whether the model's response is correct, independent of surface-level variation in phrasing.
>
> **4. Realistic gold context length.** For Gold-Context to meaningfully represent the upper bound of static RAG, the dataset must provide multi-hop oracle evidence in the form of substantive documents, not minimal snippets. Many multi-hop datasets contain extremely short oracle contexts (2–3 sentences), which trivialize the task and collapse the distinction between extraction and reasoning. In ChemKGMultiHopQA, each hop corresponds to a full paragraph (~188 tokens), comparable to retrieval scenarios in RAG systems, and requiring genuine synthesis rather than shallow extraction.
>
> In addition to the points above, the second part of our study requires that the dataset provide intermediate hops and sub-questions, enabling a mechanism-level analysis of synchronized retrieval and reasoning. This requirement further restricts the pool of usable datasets.
>
> Below is a list of recent multi-hop QA datasets (2025–2026) that we evaluated against these criteria.
>
> | Dataset | Recent & Retrieval Required | Short Answer | Gold Context Chunk Available | Scientific Domain & Corpus Based |
> |---|---|---|---|---|
> | DocHopQA | ✅ | ❌ | ✅ | ✅ |
> | BioCDQA | ✅ | ❌ | ✅ | ✅ |
> | Koblex | ✅ | ❌ | ❌ | ✅ |
> | BioHopR | ✅ | ❌ | ❌ | ✅ |
> | Cofca | ✅ | ✅ | ✅ | ❌ |
> | ChemLit-QA | ✅ | ❌ | ❌ | ✅ |
> | NovelHopQA | ✅ | ✅ | ✅ | ❌ |
> | Grade | ✅ | ✅ | ❌ | ❌ |
> | MedHopQA | ✅ | ✅ | ❌ | ✅ |
>
> (We will include this table, with cleaner formatting, in the Appendix.)
>
> Across all evaluated datasets, none simultaneously met the criteria required for controlled comparison across all three retrieval regimes and for the mechanistic diagnostics performed in Section 5.

---

> ### Author Response · Authors · 2026-03-19
> **Author Response to Reviewer ZSY4**
>
> **Justification of diagnostic metrics**
>
> We would like to clarify that the only metric relying on explicitly defined thresholds is Confidence Miscalibration. All other diagnostics—Retrieval Coverage Gap, Sufficiency Score, Anchor Carry–Drop, Composition Failure, and Distractor Latch—are either binary or continuous scores computed directly from retrieval logs, with no threshold-based decisions.
>
> For Confidence Miscalibration, the thresholds (coverage < 0.8, sufficiency < 0.6) were determined via a grid search over the joint coverage–sufficiency space. As shown in Figure 10, the accuracy landscape exhibits clear transitions around these boundaries, with sharp performance drops below either boundary. This makes the selected thresholds empirically meaningful rather than heuristic.
>
> To strengthen transparency, we have added a paragraph to the supplementary material that details the grid-search setup, evaluation grid, and selection criteria. This additional explanation should clarify the robustness of our choice and address the reviewer's concern regarding sensitivity.
>
> **Relationship to existing iterative RAG approaches**
>
> In Section 2.2, we provided an overview of existing iterative RAG approaches and how the controller design could be a practical choice based on recent works in this field. Our controller is not proposed as a novel algorithm but as a deliberately minimal diagnostic scaffold designed to instantiate the core structure shared across iterative RAG systems. As Gao et al. (2025) characterize dynamic workflow RAG, our controller implements the essential loop where "retrieval actions are conditionally triggered through continuous system introspection" — specifically via a Retrieval step, a Planning step, and an adaptive stopping decision. The Partial Answer State mechanism, which is the key design element enabling cross-step hypothesis propagation, is explicitly derived from IM-RAG (Yang et al., 2024), which we cite in Section 3.2.2. This partial answer functions as a communication channel — "explicitly stating what has been confirmed to guide subsequent retrieval" — an approach that structurally mirrors IRCoT, ITER-RETGEN, and ReSP, all of which Gao et al. categorize under the hybrid reasoning paradigm.
>
> The minimalism of our controller is intentional and diagnostic rather than a limitation. A more complex controller with learned stopping, re-ranking, or tree search would confound attribution of failures to specific mechanisms. Our goal is to isolate and measure the mechanisms themselves — retrieval coverage, anchor propagation, stopping calibration, and composition — which requires a transparent, mechanism-preserving design. We will add a paragraph to Section 3.2 making this positioning explicit.
>
> **Discussion of cost-accuracy trade-offs**
>
> In the current version, Section 5.3 and Figure 14 already provide a cost-accuracy analysis for the Iterative RAG condition, highlighting distinct model archetypes and the adaptivity–predictability trade-off. In the revision, we will extend this analysis by adding two analogous cost-accuracy plots for the No Context and Gold Context conditions, enabling a direct comparison across all three retrieval regimes. We will also clarify an important point regarding interpretation: while the Gold-Context condition represents the upper bound of static RAG performance (i.e., idealized retrieval with perfect oracle evidence), it should not be viewed as a direct surrogate for a practical static RAG system. To further strengthen this section, we have added a small ablation—requested by another reviewer—evaluating the impact of chain-of-thought reasoning in the Gold-Context setup. This ablation offers insight into how augmented reasoning within an ideal retrieval regime compares to a realistic Iterative RAG pipeline under comparable token budgets.
>
> These additions will make the performance–cost trade-offs clearer and place Iterative RAG in a more practical context relative to static RAG and its theoretical upper bound.

---

> > ### Author Response · Authors · 2026-03-19
> > **Author Response to Reviewer ZSY4**
> >
> > **Scope of the novelty claim**
> >
> > We appreciate the reviewer's comment and have refined the novelty claim to more accurately reflect prior work. As outlined in the Introduction (paragraphs 1–3), prior studies have indeed analyzed aspects of synchronized retrieval and reasoning, and several have suggested that iterative retrieval–reasoning loops could surpass static RAG. However, to the best of our knowledge, existing work typically evaluates only one of the two critical baselines:
> >
> > - *Retrieval-enhanced reasoning*, where iterative RAG is compared primarily against parametric memory (No-Context) settings.
> > - *Reasoning-enhanced retrieval*, where iterative RAG is compared against idealized evidence (Gold-Context) but usually in single-hop setups, and where oracle context is used as an upper bound.
> >
> > What has been largely missing, especially in scientific multi-hop QA, is a joint evaluation that considers both upper-bound baselines together: (1) reliance on internal parametric knowledge, and (2) performance under oracle evidence conditions in a multi-hop regime.
> >
> > Our contribution aims to fill this specific gap. We agree that the novelty claim should make this scope explicit, and we have revised the text accordingly to avoid overgeneralization and ensure clearer positioning relative to prior work.

---

### Decision · Action_Editor_y4NU · 2026-04-01

**Recommendation:** Accept with minor revision

**Audience:**

Yes

**Audience Explanation:**

Under TMLR’s acceptance criteria, the key questions are whether the claims are supported by accurate, convincing, and clear evidence, and whether at least some of TMLR’s audience would be interested in the findings. TMLR explicitly notes that novelty/significance should not be used as rejection criteria in themselves, and that uncertainty on audience interest should generally be resolved in favor of acceptance.

My judgment is that the paper clears that bar, provided the claims are kept appropriately scoped. I believe the evidence supports the following narrower claim: in the studied scientific multi-hop QA setting, and under the paper’s controlled iterative controller, staged retrieval can outperform a static oracle-evidence baseline, and the paper’s diagnostics offer useful insight into why this happens. That is a meaningful empirical finding, and one that I expect will interest a subset of TMLR readers working on RAG evaluation, multi-hop QA, and retrieval-reasoning systems.

**Claims And Evidence:**

Yes

**Claims Explanation:**

This paper studies an interesting and timely question: when, and why, can staged iterative retrieval-reasoning outperform a static oracle-evidence setup in scientific multi-hop QA? The submission evaluates 11 LLMs across three inference regimes, No Context, Gold Context, and Iterative RAG, on ChemKGMultiHopQA, and complements the accuracy comparison with a substantial diagnostic suite targeting retrieval coverage, anchor carry-drop, stopping calibration, distractor latch, and composition-related failures. Reviewers agreed that the empirical study is careful, the question is meaningful, and the diagnostic analysis is a real strength.

The main discussion centered on scope. The strongest concerns were that the paper evaluates only a single domain-specific benchmark and a single fixed-budget controller, which limits how broadly the central claim can be interpreted. One reviewer also raised a compute-fairness concern regarding Gold Context versus Iterative RAG, and questioned whether Gold Context should really be viewed as a practical static upper bound. These are legitimate concerns, and I do not think the paper supports a broad claim that iterative RAG, in general, beats ideal evidence across domains and controller families.

That said, the rebuttal materially improved the case for the paper. The authors added a compute-normalized Gold+CoT ablation, clarified that some failure analyses had been conflated in the discussion, added context-length analysis for Gold Context, clarified the thresholding story for the diagnostics, explained the controller as a deliberately minimal diagnostic scaffold rather than a claim of algorithmic novelty, and narrowed the framing relative to prior work. One reviewer explicitly indicated that their questions were addressed and updated to Leaning Accept.

For the revision needed before camera ready:
- The authors must explicitly narrow the scope of their claims to the scientific multi-hop QA domain and the specific ChemKGMultiHopQA benchmark.
- The final text should incorporate the "Gold+CoT" ablation and context-length analysis provided during the rebuttal to ensure a fair comparison of compute budgets.
- Additionally, the authors must clarify that the controller is a minimal diagnostic tool rather than a novel algorithm, fix the overlapping content in Figure 13 etc., and correct the various citation and typographical errors identified by the reviewers.

---

> ### Author Response · Authors · 2026-05-26
> **Response to Action Editor — Camera-Ready Revisions**
>
> Dear Action Editor,
>
> Thank you for the recommendation. We have addressed all requested revisions:
>
> 1. **Scope narrowed.** The abstract, introduction, discussion, and conclusion now explicitly scope our claims to scientific multi-hop QA on ChemKGMultiHopQA. Appendix S1.6 documents the dataset-selection process and acknowledges single-benchmark evaluation as a limitation.
>
> 2. **Gold+CoT ablation and context-length analysis incorporated.** The Gold+CoT ablation is now Section 4.1 (Table 2, prompt in Figure S12). The Gold Context length analysis is now Appendix S1.7 (Figures S13–S14), referenced in Section 5.3.
>
> 3. **Controller reframed as a diagnostic tool.** Sections 1, 3.2.2, and 6 now explicitly state that the controller is a minimal diagnostic scaffold rather than a novel algorithm.
>
> 4. **Figure 13 and overlapping content fixed.** Figure 13 layout adjusted, duplicated text between Sections 4.2 and 4.5 removed, and Figure 6 numbers re-aligned with the figure.
>
> 5. **Citations and typos corrected.** Thorough proofread completed.
>
> We thank you and the reviewers for the constructive feedback.
>
> Sincerely,
> The Authors